# Amorphous intermediates and discovery of a kinetic polymorph of BiVO$_4$ from heating V+Bi+Zn single-source precursors

Alexandria E. Hands[1], Thomas J. Barnes[1], Andrea Scarperi[2,3], Benjamin M. Gallant[2], Emanuele Vismara[1], Julia Wiktor[4], Stephen E. Brown[1], David Walker[5], Ashok S. Menon[6], Javier Castells-Gil[2]✉, Dominik J. Kubicki[2]✉ & Sebastian D. Pike[1]✉

Molecular single-source precursors are a promising way of obtaining multi-element extended solids directly. We show that thermal decomposition of well-defined mono-, bi- and trimetallic polyoxovanadates (POVs) proceeds through a series of intermediate amorphous and crystalline species which we characterise using solid-state NMR spectroscopy, pair-distribution function (PDF) analysis and in-situ X-ray diffraction, before forming crystalline V$_2$O$_5$ and BiVO$_4$ products. This synthetic strategy enables the formation of phases inaccessible using other routes, including a previously unknown polymorph of BiVO$_4$ which we name β-BiVO$_4$ due to its similarity to β-SnWO$_4$. Local structure information also reveals the temperature dependent incorporation of Zn dopants into BiVO$_4$. The study also explores the electrochemical properties of amorphous mixed-valence vanadium oxides as Li-ion battery electrodes. We suggest that careful analysis of the thermal decomposition of molecular species may be a way of obtaining hitherto unknown kinetically stabilised polymorphs and amorphous variants of extended solids.

Single-source precursors (SSPs) are molecules which contain all the required elements for a material. Thermal decomposition provides a convenient route to a homogeneously mixed material, potentially accessible at low temperatures and with high surface areas/porosities[1–6]. The use of SSPs can allow access to unusual phases and doping levels, or control of physical properties, that is not easily accessed by higher temperature processes[1,7–10]. Multimetallic SSPs are attractive for the synthesis of complex (nanocrystalline) multimetallic oxides, offering opportunities to control composition, as well as particle morphology[1,11–14]. A growing library of well-defined bi and trimetallic molecular precursors is available for study in this role[2,15–17].

The molecule-to-material transformation that occurs on the thermal decomposition of an SSP is a complex process requiring structural rearrangement and decomposition of any organic components. Detailed studies highlight the importance of atomic level characterisation in understanding these processes[18]. The possibility of generating interesting and useful phases at low temperatures, before the thermodynamic oxide phase, is intriguing, yet has been rarely explored in detail. Formation of amorphous phases typically makes characterisation of intermediates difficult; therefore, advanced techniques are needed to provide local structural information beyond what is available from conventional diffraction studies.

In this project the thermal transformation of mono, bi and trimetallic POV clusters is explored in the solid phase. These SSPs contain V, V + Bi or V + Bi + Zn, and allow for exploration of the intermediary stages of thermal treatment before the formation of oxide products

[1]Department of Chemistry, University of Warwick, Coventry, UK. [2]School of Chemistry, University of Birmingham, Birmingham, UK. [3]Department of Chemistry and Industrial Chemistry, University of Pisa, Via G. Moruzzi 13, Pisa, Italy. [4]Department of Physics, Chalmers University of Technology, Gothenburg, Sweden. [5]X-ray Diffraction Research Technology Platform, University of Warwick, Coventry, UK. [6]Warwick Manufacturing Group, University of Warwick, Coventry, UK. ✉e-mail: j.castells-gil@bham.ac.uk; d.j.kubicki@bham.ac.uk; sebastian.pike@warwick.ac.uk

($V_2O_5$ and (Zn-doped)-$BiVO_4$), including mixed-valence amorphous and kinetically stabilised phases.

The resultant oxide products, $V_2O_5$ and $BiVO_4$ are important materials with applications including photoelectrochemical water oxidation to enable solar-to-fuel technologies[19–23]. $V_2O_5$ is an important material, used extensively in oxidation catalysis, battery cathodes, supercapacitors, gas sensors and electro/thermo/photochromic materials[24,25]. $BiVO_4$ is a very important photoanode material with optimal band-gap properties for water oxidation[19,26], and is used in state-of-the-art artificial leaf devices[21–23]. Interest in $BiVO_4$ has rapidly accelerated in the last decade, with 860 articles published in 2024[27]. To date (in ~8000 publications), three distinct polymorphs of $BiVO_4$ have been described: the monoclinic scheelite structure (*m*-$BiVO_4$, I2/b), tetragonal zircon type structure (*tz*-$BiVO_4$, I4$_1$/amd), and tetragonal scheelite structure (*ts*-$BiVO_4$, I4$_1$/a), with *m*-$BiVO_4$ most commonly used for photoelectrochemical water oxidation. However, recent reports of high performance photoanode materials show that Mo-doping alters phase stability and promotes tetragonal scheelite[28], and, whilst the tetragonal zircon phase has a higher band-gap, it is of interest for forming type-II heterojunctions due to its high crystal symmetry[29], or, alternatively, can be used as a photocathode[30], therefore, polymorph engineering of $BiVO_4$ is of wide interest. Commonly, high-activity $BiVO_4$ photoelectrodes are produced via an electroplated BiOI phase, and may be enhanced by doping[31], however, this electrochemical method is not easily scaled whilst retaining material uniformity[19]. To deploy $BiVO_4$ photoanodes at scale, alternative synthetic routes, such as the use of SSPs[15], require exploration.

POV clusters make attractive SSPs, as they are typically anionic, they co-crystallise with a cation, enabling two or three different metals to be incorporated homogeneously into a crystalline structure. The thermal decomposition of V(V) decavanadate ([$V_{10}O_{28}$]$^{6-}$) compounds was explored by Ulická, Baran and co-workers to make ternary oxide materials (including mixtures), and, more recently, Streb and co-workers considered the impact of thermal treatment when preparing battery electrodes from POVs[32–37]. Initial endothermic processes result in dehydration and molecular restructuring[36,38]. In most cases partial reduction of V(V) to V(IV) occurs at moderate temperatures (maximum reduction at ~200 °C)[34,35]. Partial re-oxidation occurs on increasing annealing temperature, as the final oxide phases crystallise, however, electron paramagnetic resonance (EPR) spectroscopy indicates that some oxygen vacancies may remain[38]. The thermal decomposition of ammonium metavanadate [$NH_4$][$VO_3$] also accesses nanostructures of $V_2O_5$. Initial restructuring to [$NH_4$]$_2$[$V_6O_{16}$] is followed by intermediary, partially-reduced, black phase at 150 °C–200 °C, before $V_2O_5$ formation at 250 °C[39,40]. The release of $NH_3$ and $H_2O$ gasses aids construction of mesoporous structures[41].

No decavanadate structure with an associated bismuth counter-cation is known, instead, reaction of decavanadate with Bi($NO_3$)$_3$ results in the neutral compound [$H_3Bi_4V_{13}O_{40}$(DMSO)$_{12}$]·4(DMSO) (**2**), in which a [ε-($VO_4$)$V_{12}O_{36}$]$^{15-}$ Keggin anion core is capped by Bi cations that are supported by DMSO ligands[42]. Altering the synthesis by the addition of chloride ions, allows the formation of the [$Bi_2V_{12}O_{33}$Cl(DMSO)$_6$]$^-$ anion which may be partnered with a range of metal cations ([$Bi_2V_{12}O_{33}$Cl(DMSO)$_6$]$_x$[M(DMSO)$_y$]$_z$, **3-M**)[15,16,43]. These POVs have been used as SSPs for $BiVO_4$ thin-films; **2** or **3-M** (M = Co, Ni, Cu, Zn) were dropcast onto a fluorine-doped tin oxide (FTO) substrate and annealed at 550 °C to generate (M-doped) $BiVO_4$ photoanodes on up to a 300 cm$^2$ scale[15]. It is noteworthy that the V:Bi ratio in **2** and **3** is 3.25 and 6 respectively, larger than required for $BiVO_4$, and, therefore, thermal decomposition also forms a $V_2O_5$ byproduct. However, there are advantages of a V rich precursor:

- A V-rich environment ensures that no Bi rich phases are formed, noting that in chemical vapour deposition approaches to $BiVO_4$, $Bi_2VO_5$ is easily produced as a by-product, likely due to loss of some V as volatile species in the process[44].

- Recent reports indicate that heterojunctions between $BiVO_4$ and $V_2O_5$ can be advantageous for photocatalysis[45].
- The $V_2O_5$ by-product in the decomposition of **2** or **3-M** is easily removed by a simple basic washing step, which aids creating a porous $BiVO_4$ architecture[15]. Furthermore, deliberately releasing V ions into the electrolyte can be advantageous for retarding photodegradation of $BiVO_4$[46].

Of the doped $BiVO_4$ photoanodes produced from **3-M**, the Zn-doped material showed the highest photocurrent density for water oxidation, however, the role of the Zn atoms was not clear[15]. Scanning electron microscopy with energy-dispersive X-ray spectroscopy (SEM-EDS) data suggested 1.5 Zn$^{2+}$ dopants replace Bi$^{3+}$ sites for 9% of the Bi atoms. Subsequent studies, using alternative synthetic routes, have shown that Zn$^{2+}$ dopants can replace Bi$^{3+}$ sites, but in these cases there are associated oxygen vacancies. These studies suggest that Zn-doping promotes absorption of water to Bi sites[47]. Zn doping may also increase n-type conductivity; improve the number of oxygen chemisorption sites that enhance surface charge transfer in photoelectrocatalysis; enhance surface reaction kinetics; and can be useful for forming n-n$^+$ heterojunctions (e.g., with Mo-doped $BiVO_4$)[47–49].

In this study the thermal decomposition of [$NH_4$]$_6$[$V_{10}O_{28}$]·6$H_2O$ (**1**), [$H_3Bi_4V_{13}O_{40}$(DMSO)$_{12}$]·4(DMSO) (**2**) and [$Bi_2V_{12}O_{33}$Cl(DMSO)$_6$]$_2$ [Zn(DMSO)$_6$] (**3-Zn**) are explored, with a focus on understanding intermediate phases that occur during these molecule-to-material transformations (Fig. 1).

## Results and discussion

### Thermal decomposition of ammonium decavanadate (1)

With the ultimate aim of studying **2** and **3-Zn**[15], a monometallic cluster, **1**, with a similar size and structure was chosen as an entry point to the study. While **1** contains the well-studied decavanadate anion[50], to the best of our knowledge, the thermal transformations of the ammonium salt have only been briefly discussed[51]. Therefore, we set out to identify any intermediate stages, including amorphous ones, using X-ray diffraction, PDF and solid-state NMR analysis. Bright orange **1** was prepared as a powder from literature routes[52], and its structure confirmed by IR spectroscopy (Fig. S1), H, N elemental analysis, and single crystal and powder X-ray diffraction (PXRD, Fig. S2)[53]. On drying under extended vacuum around half of the co-crystallised water is lost. $^{51}$V MAS NMR of **1** shows multiple V environments in the ranges of −405 to −439 ppm and −480 to −555 ppm (Fig. 2a), consistent with solution spectra[54], and with full-width-half-maxima (FWHM) of 4–5 ppm. The spectrum implies that there are several different decavanadate environments in this microcrystalline material, likely due to local hydration levels or partial protonation.

Thermogravimetric analysis (TGA-MS/DSC) shows several consecutive mass losses totalling ~24% mass loss on heating to 370 °C, with water and ammonia detected by mass spectrometry throughout, but with enhanced water loss between 80 °C–150 °C (Figs. S3–5). A mass loss of 22.5% would be expected based on the loss of all $NH_3$ and $H_2O$ from **1** and formation of $V_2O_5$, therefore, some partial reduction (and loss of oxide, noting that some NO loss is observed) and/or loss of volatile V compounds may occur during this process. Figures 2b and S6–7 show X-ray diffraction patterns of **1** heated in-situ from RT to 600 °C. The initial crystalline structure is maintained until 80 °C before a new broadened diffraction pattern is observed, consistent with a dehydration step[35,36,38,53]. Infra-red spectroscopy of a sample heated to 150 °C–175 °C shows a shift and splitting of the terminal V=O stretching frequency from ~937 cm$^{-1}$ to ~958 cm$^{-1}$ and a minor signal at 995 cm$^{-1}$ (Fig. S8) which could imply restructuring of decavanadate into hexavanadate, [$NH_4$]$_2$[$V_6O_{16}$][55]. H, N analysis of a sample heated to 150 °C is a close match for hexavanadate with a small amount of remaining decavanadate (Table S2), A poorly crystalline unidentified phase is observed at 160 °C by PXRD (Fig. S9), although this pattern

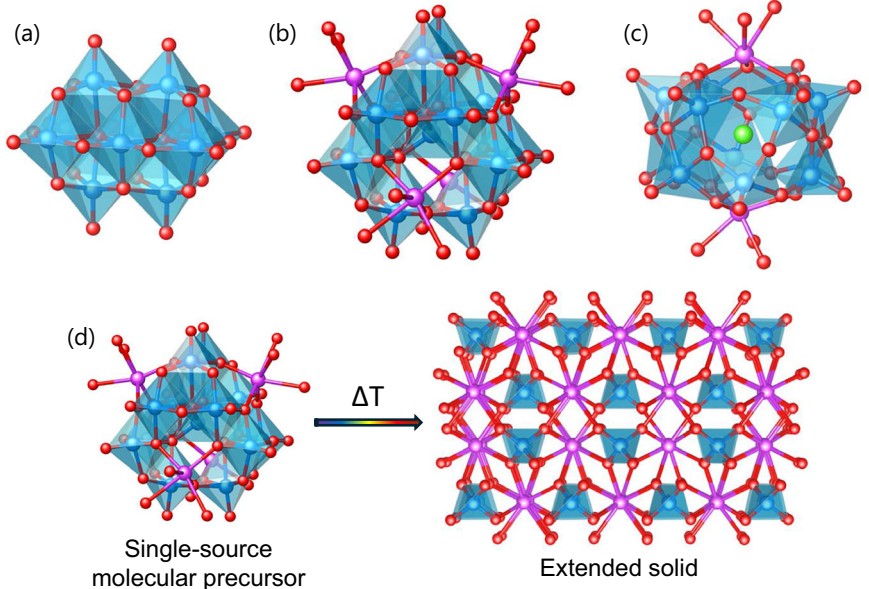

**Fig. 1 | Metal-oxygen connectivity of the molecular precursors. a–c** Crystal structures of **1**, **2** and **3** (anion only), SMe$_2$ groups of DMSO ligands coordinated to Bi, and all H atoms omitted for clarity. **d** Scheme of the strategy used to synthesise *m*-BiVO$_4$ from well-defined molecular precursors (V$_2$O$_5$ by-product not shown). Colour code Bi, purple; V, blue; O, red; and Cl, green.

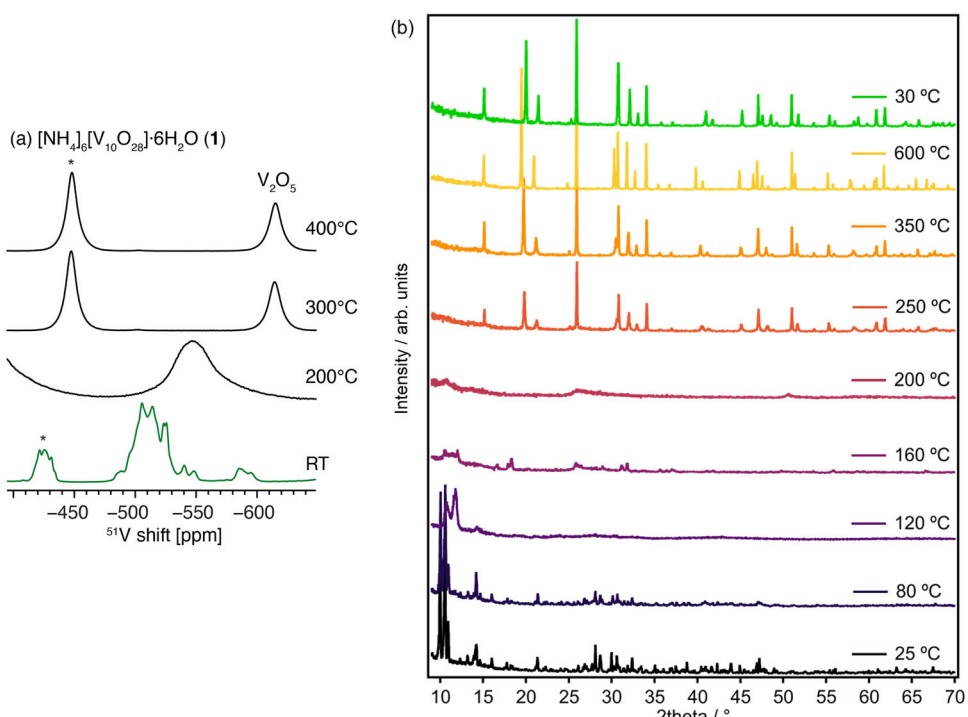

**Fig. 2 | Thermal transformation of 1. a** Room-temperature $^{51}$V solid-state MAS NMR spectra of **1** and the products of thermal transformations. The temperatures given correspond to the maximum heating temperature. The asterisks indicate spinning sidebands. **b** Variable temperature PXRD of compound **1** and the products of heating.

does not match previously reported [NH$_4$]$_2$[V$_6$O$_{16}$] (Fig. S10). Further heating results in a loss of crystallinity at 200 °C. By 250 °C α-V$_2$O$_5$ signals are emerging in the PXRD (and also IR spectra), and these sharpen on further heating (Figs. 2b, S6–8 & Table S3). The crystallite size was monitored using the double-Voigt methodology reported by Balzar et al.[56]; powdered **1** was estimated to have average crystallite sizes of ~10 nm at room temperature, whilst, after heating to 250 °C, V$_2$O$_5$ crystallites are ~103 nm and continue to grow to ~162 nm on reaching 600 °C (~160 nm estimated after cooling).

Samples of **1** were annealed ex-situ to different maximum temperatures for more in-depth analysis at each stage. The samples change from orange to brown (after 150 °C) to black (after 200 °C, **1$^{200 °C}$**) before turning red/brown after 300 °C (Fig. S11). The black phase **1$^{200 °C}$** is notable, especially considering that **1** does not contain any carbon. $^{51}$V MAS NMR spectroscopy of **1$^{200 °C}$** revealed one broad vanadium environment at −550 ppm (FWHM of ca. 40 ppm, i.e., ten times broader than in pristine **1**), consistent with an amorphous phase (Fig. 2a). After 300 °C, a single V environment is observed at −615 ppm

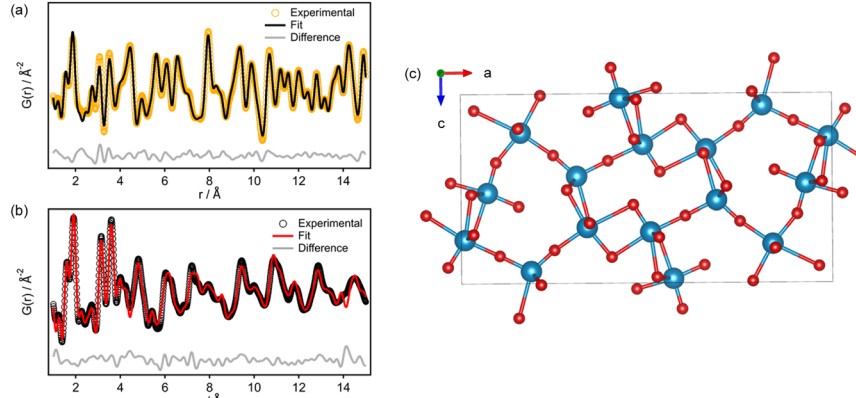

**Fig. 3 | X-ray PDF data of 1 after annealing and structure of $V_4O_9$.** X-ray PDF fits ($Q_{max}$ = 22.0 Å⁻¹) of the materials obtained after annealing **1** at **a** 400 °C ($V_2O_5$, Rw = 12.2 %), and **b** 200 °C (Refined weight fractions: $V_4O_9$: 50(5) wt. %; $(NH_4)V_4O_{10}$: 35(4) wt. %; **1**: 15(7) wt. %, Rw = 14.8 %). **c** Snapshot of the $V_4O_9$ structure (*Pnma*) viewed along the *b* axis. Colour code: V, blue; O, red.

(FWHM of 12 ppm), consistent with $V_2O_5$, and there are no further spectral changes after 400 °C.

X-ray pair distribution function analysis was conducted on **1**, amorphous **1²⁰⁰ °C** as well as a sample heated to 400 °C (Fig. S12). The sample heated to 400 °C shows an excellent match for $V_2O_5$ (Fig. 3a). **1²⁰⁰ °C** reveals a different PDF pattern, which was found to resemble $V_4O_9$ (Pnma[57,58], Rw = 30.4% after refining thermal factors and atomic positions, Fig. S13, Table S4). In Pnma $V_4O_9$ half the vanadium atoms have an oxidation state of IV, a quarter of sites are 6-coordinate with the remainder 5-coordinate (square based pyramidal, Fig. 3c). The fit was initially improved by adding a contribution from either decavanadate or hexavanadate anions (Rw = 20.1%, (**1**) = 21.4 wt%, Fig. S13, or Rw = 26.7%, ($[NH_4]_2[V_6O_{16}]$) = 13.2 wt% respectively), which may be present in the material at this stage of the thermal transformation, but without long-range molecular packing that would give diffraction signals. The fit was improved even further by incorporating some ammonium vanadium bronze $(NH_4)V_4O_{10}$ (noting similar phases have been previously observed from hydrothermal heating of $[NMe_4]_4[H_2V_{10}O_{28}]$); giving a two-component fit with Rw = 17.8%[54,59,60]. The combination of $V_4O_9$, $(NH_4)V_4O_{10}$ and **1** resulted in the best fit to the PDF with Rw = 14.8% and relative phase weight fractions $V_4O_9$:$(NH_4)V_4O_{10}$:**1** 50(5):35(4):15(7) (Fig. 3b, S14). No improvement in the fit occurred by adding $V_2O_5$. This suggests that **1** undergoes a series of transformations during thermal decomposition to form mixed-valence $V_4O_9$ and $(NH_4)V_4O_{10}$ before its full conversion to $V_2O_5$ beyond 250 °C. From this mixture of materials, ~35% of V might be expected in the V(IV) oxidation state and the presence of V(IV) was confirmed by EPR (Fig. S15). However, **1²⁰⁰ °C** was dissolved in 2 M $H_2SO_4$ and a double titration method used to identify the amount of V(IV) content[61], and a lower value of 2.7 ± 1 % V(IV) was identified. X-ray Absorption Near Edge Structure (XANES) analysis confirmed that V(V) is dominant as the spectra of $V_2O_5$, **1** and **1²⁰⁰ °C** appear almost identical, (noting that XANES is less sensitive to the presence of V(IV) than EPR spectroscopy, Fig. S16). This low V(IV) content is consistent with previous studies[35], yet is surprising considering the black colour of **1²⁰⁰ °C**. The colour must arise from efficient d-d transitions and intervalence charge transfer (IVCT) excitations that occur across the visible range[25].

The reduction of some V at intermediary stages of heating is caused by a redox reaction with ammonia evolved from **1**, in fact, treating $V_2O_5$ with $NH_3$ is a synthetic route to $V_4O_9$[62]. Partial reduction is consistent with literature reports[32,34,35,39], but here we extend on these by identifying a likely structure of this initial amorphous phase. In **1²⁰⁰ °C**, the local connectivity resembles a mixture of $V_4O_9$, and $(NH_4)V_4O_{10}$ but the low V(IV) content implies that most sites are occupied by V(V), implying that these may be oxygen-rich kinetically stabilised phases.

¹H MAS NMR of **1** reveals multiple proton environments at 9–14 ppm, attributed to strongly coordinated $H_2O$ and $[NH_4]^+$ owing to their substantial deshielding compared to bulk water (ca. 5 ppm) and $[NH_4]^+$ in aqueous solutions (ca. 7 ppm), noting that their chemical shifts strongly depend on the pH and hydrogen bonding network. After heating to 200 °C, all these species are lost, and a new peak is found at 7.5 ppm, accompanied by a substantial decrease in the signal-to-noise ratio (SNR) of the spectrum (20% of that in pristine **1**), consistent with the loss of the volatile proton-containing species (Fig. S17a). H, N elemental analysis collected after annealing to 200 °C and 250 °C (Table S2), and FT-IR spectroscopy (Fig. S9) confirm the loss of most H and N over this temperature range, in good agreement with the estimated phases from PDF fitting. At 250 °C, once the PXRD reveals $V_2O_5$ is forming, only trace ammonium remains, and by 300 °C only residual protons are observed in the ¹H MAS NMR with an SNR of <1% of that observed in pristine compound **1** (Fig. S17a).

In summary, upon heating **1**, an initial dehydration process occurs, resulting in collapse of the crystalline structure, further restructuring of the decavanadate structure via partially crystalline phases occurs before complete loss of crystallinity at 200 °C, at which stage the sample becomes black, with ~3% V(IV) content, and is best described with a local structure resembling a mixture of Pnma $V_4O_9$ and layered $(NH_4)V_4O_{10}$ and with some remaining isolated POV anions. Heating beyond this point begins to form $V_2O_5$ with the chemical transformation completed by 400 °C. Further heating under air reduces the oxygen vacancy content and increases the crystallinity (Fig. 4).

## Thermal decomposition of 2 and 3

**2**, $[H_3Bi_4V_{13}O_{40}(DMSO)_{12}]\cdot4(DMSO)$, and **3-Zn**, $[Bi_2V_{12}O_{33}Cl(DMSO)_6]_2[Zn(DMSO)_6]\cdot12(DMSO)$ (N.B. co-crystallised DMSO content drops if dried under vacuum), were prepared by literature methods and the expected structures confirmed by PXRD (Figs. S18–19, see supporting note 1 for discussion of how to avoid possible impurities, including $[Bi_2V_{12}O_{33}Cl(DMSO)_6]_2[VO(DMSO)_5]\cdot8(DMSO)$, **3-VO**). It is noteworthy that precipitation of **2** may generate an amorphous form (**2ᵃᵐᵒʳ**) which contains the expected C, H, S %, whilst slower crystallisation generates a (micro)crystalline form (**2**).

The crystal structure of **2** contains four distinct V sites. ⁵¹V MAS NMR of a powdered sample of **2** reveals multiple V environments between −440 and −580 ppm (Fig. 5a, RT). The spectrum is markedly broadened by static disorder owing to the presence of co-crystallised

DMSO molecules. Both cross-polarisation and direct excitation $^{13}C$ MAS NMR lead to fast signal build-up and a comparable SNR, which indicates that most DMSO molecules in the structure of **2** are rigid (i.e., there is static disorder) rather than fluxional (Fig. S20). This may impact the ability of DMSO to escape the crystals and promote decomposition rather than evaporation on heating.

TGA under air shows expected mass losses (**2$^{amor}$**, 35.5%; **3-Zn**, 40.2%) during heating as DMSO is lost via decomposition (and water is also lost from **2**, Fig. S21-26)[15,42]. A curious mass increase is seen when heating **2$^{amor}$** under air (5 °C/minute) between 360 °C and 450 °C (Fig. S21), associated with an exothermic process (Fig. S22). This additional mass is then lost again upon heating to 520 °C. The mass gain may be due to oxidation reactions occurring to V or S atoms within the structure at this stage. A temperature of 520 °C or 600 °C is required to complete the mass loss and formation of the extended oxide materials (*m*-BiVO$_4$ and V$_2$O$_5$, after cooling to room temperature) for **2$^{amor}$** and **3-Zn** respectively. TGA-MS implies DMSO decomposition occurs, with indication of CO$_2$ and H$_2$CO, and some evidence of SO$_2$ and/or H$_2$S loss occurring (Figs. S23 & S26). **2/2$^{amor}$** & **3-Zn** are red, however, after

heating above 175 °C (under air) the resulting materials appear black, before becoming green (400 °C–480 °C) and then converting to the final orange/yellow colour at ~550 °C (Fig. S27-28). The dark colours are associated with V(IV) content due to d-d and IVCT transitions and are consistent with the thermal transformation of **1**. The $^1H$ MAS NMR spectrum of **2** heated at 300 °C shows the presence of intense spinning sidebands, not present at any other temperature, which we attribute to paramagnetic effects (anisotropic bulk magnetic susceptibility and/or pseudocontact shifts, Fig. S17d).

C, H, S elemental analysis, shows that **2$^{amor}$** heated to 200 °C loses DMSO (via decomposition) but retains a higher-than-expected H content, likely indicating retention of hydroxide functionality (Table S5). $^1H$ MAS NMR of this material shows the presence of multiple sharp, liquid-like species, across the spectrum, including in the aromatic (7-9 ppm) region. Since the only solvent used in the synthesis of **2** was DMSO, we cautiously attribute these species to DMSO cyclisation products catalysed by the oxide phases. FTIR analysis reveals loss of all DMSO after heating to 400 °C (Fig. S29), corroborated by the C% content and $^1H$ and $^{13}C$ MAS NMR which confirm the loss of the DMSO signal ($^1H$, 3.5 ppm, and $^{13}C$, 40.5 ppm) and the generation of a broad, −5 to +10 ppm $^1H$ signal after heating to 600 °C (Figs. S17, S20), which we attribute to residual protonation of the oxide surface sites (SNR of <0.1% of that in pristine **2**). Some H and S content are retained at 420 °C, indicating some incorporation of S into the oxide materials at this stage (Table S5), but after 600 °C essentially all C, H and S are lost. **3-Zn** also retains higher-than-expected H & S content during heating, and after 380 °C the sample still contains 1 wt% S (Table S6). $^1H$ MAS NMR of **3-Zn** also reveals loss of DMSO signals and, first, generation of intermediate proton-containing degradation products, and then residual proton-containing species (SNR < 1% of that in pristine **3-Zn**) after heating at 600 °C (Fig. S17c).

PXRD was collected on samples of **2** and **3-Zn** heated ex-situ (samples pre-heated and cooled, Fig. S30) and on samples of **2$^{amor}$** and **3-Zn** heated in-situ (Fig. 6, S31–32). The in-situ samples were pre-heated to 175 °C ex-situ to remove the majority of DMSO in order to protect the equipment. **3-Zn** shows restructuring of the crystalline phase after 60 °C, giving unknown phase(s) likely to retain POV species

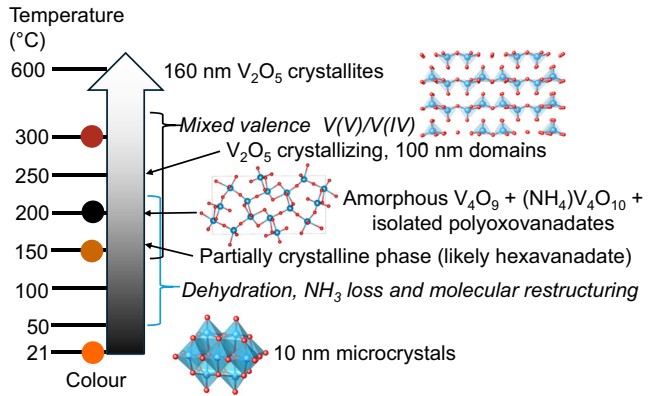

**Fig. 4 | Summary diagram of the molecule-to-material thermal transformation of 1 into V$_2$O$_5$.** Colour code in structures: V, blue; O, red.

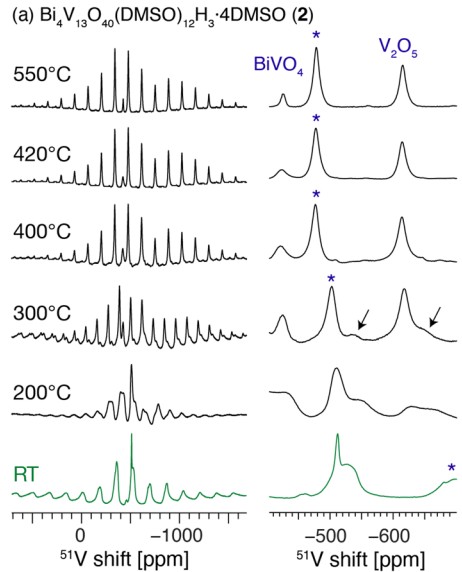

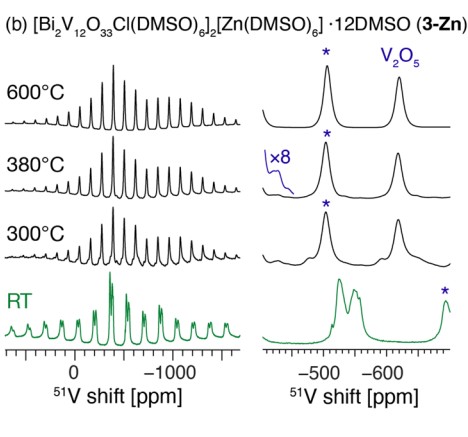

**Fig. 5 | Solid-state NMR characterisation of the thermal transformations of 2 and 3-Zn.** Room-temperature $^{51}V$ solid-state MAS NMR spectra of the starting materials and the products of their thermal transformations (ppm = parts per million). **a** compound **2**, **b** compound **3-Zn**. The temperatures given correspond to the maximum heating temperature. The asterisks indicate spinning sidebands. The

signals attributed to β-BiVO$_4$ are indicated by arrows, which indicate the possible positions of the isotropic resonance and a spinning sideband. While β-BiVO$_4$ is expected to contain only one distinct V site, the large linewidth makes the identification of the isotropic peak challenging.

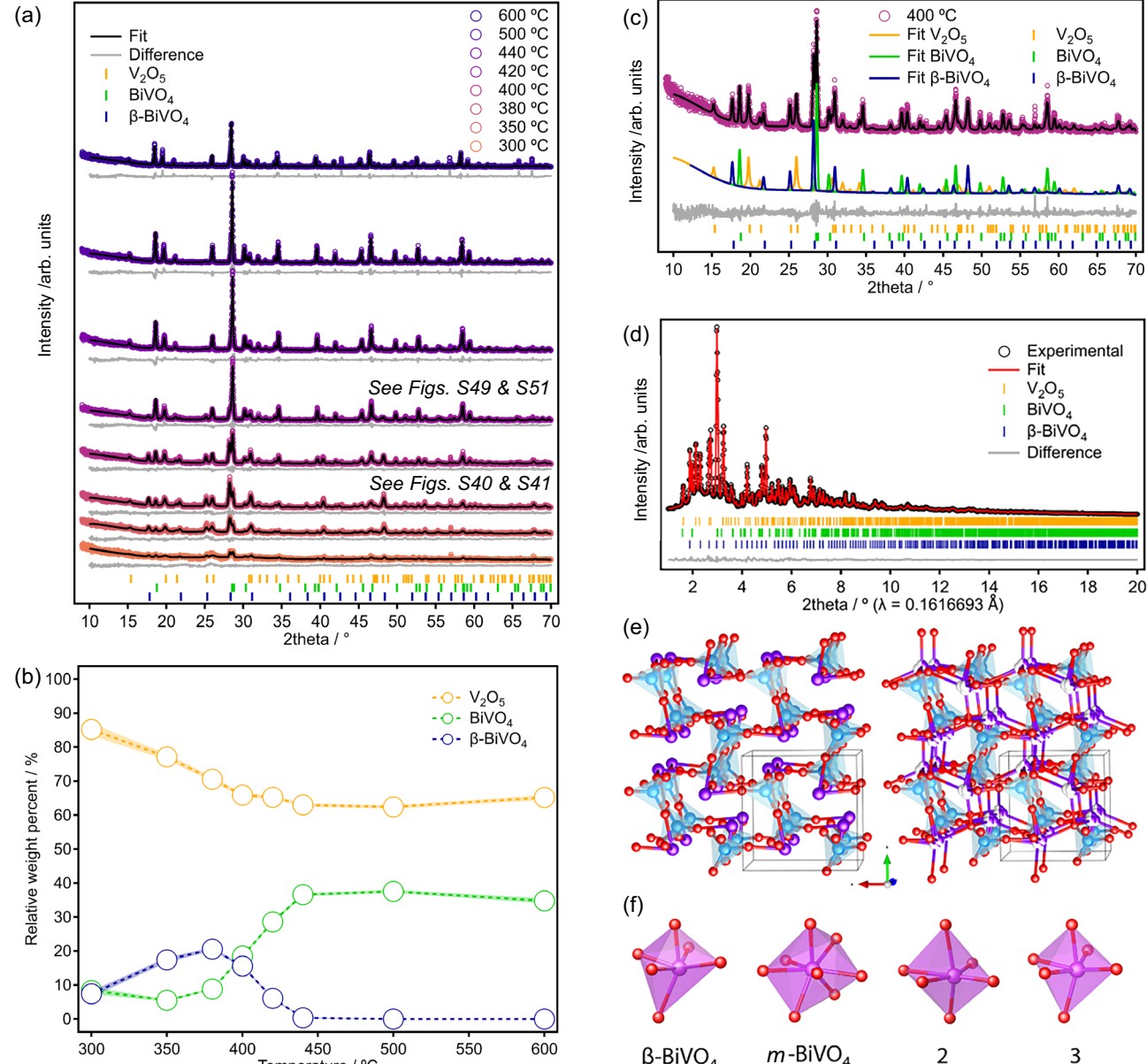

**Fig. 6 | Characterisation and structure of β-BiVO₄ during thermal transformations of 3-Zn. a** Variable temperature PXRD patterns starting from **3-Zn** (sample pre-heated to 60 °C, cooled, then heated to 175 °C and cooled before VT-experiment 200 °C–600 °C, then cooled to 30 °C), alongside an inset (**c**) of the fit at 400 °C showing the contributions of the three different phases. **b** Evolution of the relative percent of each crystalline phase during annealing of **3-Zn** as obtained from the Rietveld analysis of VT-PXRD data of **3-Zn**. **d** Rietveld fit of **3-Zn** annealed to 380 °C then cooled (data collected at Diamond Light Source, I15-1, *See ESI for details*) after introducing Bi disorder. Refinement details: V₂O₅ (63.1(2) wt. %),

*Orthorhombic (Pmmn), a = 11.509(1) Å, 3.5675(2) Å, c = 4.3795(5) Å; m-BiVO₄ (11.8(1) wt. %), Monoclinic (I2/b), a = 5.1500(8) Å, b = 5.1110(6) Å, c = 11.722(2) Å, β = 89.98(2)°; β-BiVO₄ (25.1(2) wt. %), Cubic (P2₁3), a = 6.9916(2) Å. Rwp = 2.26 %, Rexp = 1.45, X2 = 1.59.)* after introducing Bi disorder. **e** View of the refined β-BiVO₄ structure without (left) and with (right) disordered Bi atoms, disordered atoms each with an occupancy of 1/3. **f** Comparison of the Bi–O coordination environment in β-BiVO₄ with those observed in m-BiVO₄ (8-coordinated) and in compounds **2** and **3** (both 6-coordinated). Colour code: Bi, purple; V, light blue; O, red.

before the formation of a mainly amorphous phase at 175 °C (Fig. S19). Starting from either **2ᵃᵐᵒʳ** or **3-Zn** an amorphous phase is found at 200 °C, before signals for BiVO₄ and V₂O₅ appear after 300 °C and become sharper upon heating further. N.B. In the in-situ heated samples the high-temperature tetragonal scheelite phase, ts-BiVO₄, is observed during annealing up to 600 °C but this phase transitions to the stable monoclinic form, m-BiVO₄, when cooled to room temperature.

In samples of in-situ heated **3-Zn** and ex-situ heated (microcrystalline) **2** an extra phase is observed between 350 °C–420 °C, which is then lost on further heating (Fig. 6, S30). Comparing this

phase with all reported Bi, V and Bi-V oxides in the database via Pawley and Rietveld refinements yielded unsuccessful results. Indexing of the unit cell with the softwares DICVOL[63] and TREOR[64], however, suggested a primitive cubic unit cell for the unidentified phase with cell parameter a ~ 7.03 Å. This kinetic phase (sample heated to 400°C), was investigated further, and was found to resemble the β-SnWO₄ phase[65], which is also found in the family of oxide-conducting Ln₂Mo₂O₉ (Ln = La, Pr) LAMOX compounds[66–68]. Therefore, we carried out a Rietveld refinement (See ESI, Tables S7–S8, Figs. S33–41, for details) on synchrotron X-ray data collected in I15-1 at Diamond Light Source using the β-SnWO₄ structural model but replacing Sn²⁺ by Bi³⁺ and W⁶⁺

by $V^{5+}$[65]. This gave excellent results, confirming that this new kinetic phase crystallises in the cubic space group $P2_13$ with cell parameter $a = 6.9916(2)$ Å with Bi and V atoms located on the 3-fold symmetry axis of the cubic unit cell. Therefore, we dub this new phase β-BiVO$_4$. From the unit cells, the β phase is expected to be less dense (6.3 g/cm³) than $m$-BiVO$_4$ (6.8 g/cm³). The structure of β-BiVO$_4$ (Fig. 6e, S42–43) consists of [VO$_4$]³⁻ tetrahedra with an average V–O bond distance of 1.62(2) Å and bond angles ranging between 102.4-115.5°. The [VO$_4$]³⁻ tetrahedra are directly coordinated to six different Bi³⁺ ions, which display a very distorted BiO$_6$ octahedral environment (Fig. 6e, f, S42–43). In this regard, it can be viewed as a rock salt-type structure with alternating Bi³⁺ and [VO$_4$]³⁻ units. Similarly to Sn²⁺ in β-SnWO$_4$, Bi³⁺ ions display three short (2.26(1) Å) and three long (2.662(5) Å) Bi–O distances, due to the presence of the lone electron pair, in keeping with previous reports that indicate stabilisation of Bi³⁺ coordination environments in BiVO$_4$ structures[65,69]. This behaviour is reminiscent of β-SnWO$_4$, even though the lone pair in Bi³⁺ is less localised than in the case of Sn²⁺, which would explain the preference to form $m$-BiVO$_4$ over β-BiVO$_4$[69]. Similarly to in β-SnWO$_4$, the Bi³⁺ lone pair was found to be oriented towards the base of the [VO$_4$]³⁻ tetrahedra. However, the large $B_{iso}$ value for Bi with respect to V and O, suggests that Bi ions may be disordered. Bi was allowed to refine outside the special position, resulting in a slight improvement of the fit as well as for the $B_{iso}$ value for Bi, strongly suggesting that Bi atoms slightly deviate from the 3-fold axis into three closely spaced positions (Fig. 6e, S41, S43). Similar disorder has been found for Ln and Mo ions in structurally-related Ln$_2$Mo$_2$O$_9$ (Ln = La, Pr) oxide conductors[68], and could have important implications in the photocatalytic or ionic conductivity properties of this β-BiVO$_4$ phase. It is noteworthy that the (6-coordinate) Bi environment in β-BiVO$_4$ is similar to in the precursors **2** or **3-Zn**, but is rather different to the (8-coordinate) environment found in monoclinic or tetragonal BiVO$_4$, and exists with a wider range of Bi–O distances (2.26-2.66 Å $vs$ 2.43-2.52 Å in tetragonal BiVO$_4$, Fig. 6f)[26]. The V environment also exhibits shorter V–O bond lengths compared to other polymorphs (1.62 Å compared to 1.69-1.77 Å)[26]. Recent studies highlight how judicious choice of precursors can access metastable phases during solid-state synthesis[70]. S-doping of the BiVO$_4$ phase may be influential on the formation and stability of β-BiVO$_4$; S-doping has been studied in the related La$_2$Mo$_2$O$_9$ phase[71], but is also known in monoclinic BiVO$_4$[72].

DFT calculations were performed to examine the β-BiVO$_4$ phase and compare it to the thermodynamically stable $m$-BiVO$_4$ phase. Computational details are given in the supporting information (Figs. S44–46, Tables S9–10). The β-BiVO$_4$ structure is found to be higher in energy than $m$-BiVO$_4$ by 19.5 meV per atom, consistent with its kinetic origin. Despite the close chemical composition and identical oxidation states, the two phases exhibit significant electronic differences. An important increase in the band gap is observed in β-BiVO$_4$, amounting to +1.09 eV within the r²SCAN functional and to +1.58 eV within more advanced non-empirical hybrid functionals. Figure 7 shows the density of states (DOS) for the two phases, aligned through the O $2s$ levels to allow direct comparison. The band-gap opening arises from shifts in both the valence-band maximum (VBM) and the conduction-band minimum (CBM).

We attribute the larger band gap in β-BiVO$_4$ to structural differences between the two phases. β-BiVO$_4$ exhibits more widely separated VO$_4$ tetrahedra (shortest V-V ≈ 4.39 Å vs 3.88 Å in $m$-BiVO$_4$), leading to weaker V 3d-O 2p-V 3d interactions and a narrower, more localised conduction band (Figs. S44-45). At the same time, the Bi coordination changes from nearly uniform 8-fold Bi-O bonding in the monoclinic phase to a strongly split 6-fold environment, which reduces Bi 6s-O 2p mixing at the valence-band maximum. Together, these effects raise the CBM and lower the VBM, consistent with the larger band gap in β-BiVO$_4$. Excitonic effects were evaluated using time-dependent hybrid DFT (Fig. S46). Both phases exhibit comparable

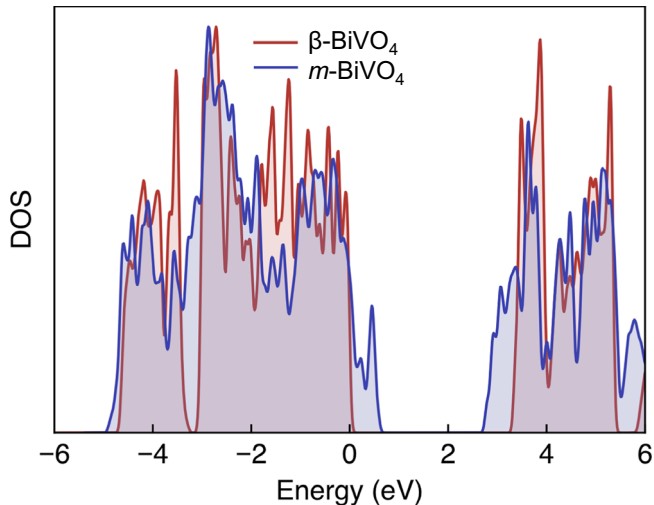

**Fig. 7 | Comparison of density of states (DOS) for the β-BiVO$_4$ and $m$-BiVO$_4$ phases.** Calculations were performed within the r²SCAN functional and aligned through the O $2s$ states.

exciton binding energies (≈ 0.4–0.5 eV). These values are most likely overestimated[73], but the similarity between the strength of the effect should be well captured. We note that absolute band-gap values in BiVO$_4$ depend sensitively on the level of theory and on additional physical effects such as electron–phonon coupling[74]. Further tests are provided in the SI.

Rietveld analysis of VT-PXRD data collected between RT-600 °C reveals the evolution of the relative amounts of oxide phases; **2ᵃᵐᵒʳ** primarily yields $ts$-BiVO$_4$ at 300 °C followed by the subsequent crystallisation of V$_2$O$_5$ as the temperature increases (Fig. S32), in contrast, for microcrystalline **3-Zn**, V$_2$O$_5$ appears as the major species at 300 °C before BiVO$_4$ phases grow in (Fig. 6b), perhaps the relatively higher temperature crystallisation of BiVO$_4$ is due to the added influence of Zn atoms on the BiVO$_4$ phases. From **3-Zn**, the β-BiVO$_4$ phase is the dominant Bi containing phase <400 °C, however, from 440 °C, only V$_2$O$_5$ and $ts$-BiVO$_4$ are observed. After annealing **2ᵃᵐᵒʳ** to 600°C V$_2$O$_5$ and $ts$-BiVO$_4$ appear in a 50.1(4): 49.9(4) wt. % ratio. Heating **3-Zn** to the same annealing temperature gives 66.0(7): 34.0(7) wt. % ratio of V$_2$O$_5$ and $ts$-BiVO$_4$. Only smaller quantities (~2.5 wt%) of kinetic phase β-BiVO$_4$ could be identified after in-situ heating of **2ᵃᵐᵒʳ** (compared to up to 40% for ex-situ heating of microcrystalline **2**) – it is possible that the crystallinity of the precursor influences the kinetics of the molecule-to-material transformation (Fig. S47).

Analysis of the crystallite size (starting from **2ᵃᵐᵒʳ**) shows an average crystallite size of 18 nm for BiVO$_4$ and 5 nm for V$_2$O$_5$ at 300 °C (which is much smaller than observed during similar processing of monometallic **1**). Crystallite size then increases with temperature, reaching 133 nm and 161 nm, respectively, at 600 °C.

Samples of **2** and **3-Zn** heated to 400 °C or 420 °C were analysed by X-ray PDF (at room temperature). The XRD patterns gave a good fit for a mixture of $m$-BiVO$_4$, β-BiVO$_4$ and V$_2$O$_5$ as expected (Fig. S35–38, S48–51). For **2** after 400 °C, PDF analysis suggests 43(6) wt% is β-BiVO$_4$, however this drops significantly by 420 °C, at which point weight fractions are 43(4)% V$_2$O$_5$: 53(4)% BiVO$_4$: 4(2)% β-BiVO$_4$. The weight fractions from **2** and **3-Zn** after 420 °C are broadly in line with the expected quantities of oxides from the precursor formula (Table S11), however, after annealing to 600°C ICP-MS of the final material shows the amount of V has decreased, which must be due to release of volatile V species (Table S12)[75]. In the previously reported preparation of thin-films of BiVO$_4$ from **2**, no crystalline V$_2$O$_5$ phase was observed, attributed to amorphous

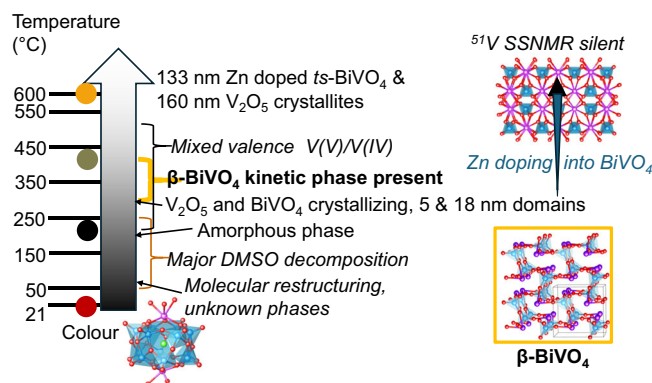

**Fig. 8 | Summary diagram of the molecule-to-material thermal transformation of 2 or 3-Zn into (Zn-doped) BiVO₄ and V₂O₅.** SSNMR = solid-state nuclear magnetic resonance. Colour code in structures: Bi, purple; V, blue; O, red; and Cl, green.

$V_2O_5$, but perhaps this is better explained through loss of volatile $V$[15].

Our refinement results also showed slightly different cell parameters for $m$-BiVO₄ from **2** and **3-Zn** heated to 420 °C. In the Zn-doped $m$-BiVO₄ phase (from **3-Zn**) there is a slight increase of -0.015 Å along the $a$-axis and a very slight decrease in the $b$ and $c$-axes, with respect to $m$-BiVO₄ from **2** (Figs. S34 and S49). This is accompanied by a lower relative occupancy of 0.945(6) at the Bi site in the Zn-doped phase (compared to 0.985(4) when from **2**). The refinement of Bi site occupancy also slightly improved the refinement of **3-Zn** from $R_{wp}$ = 2.30 % to $R_{wp}$ = 2.26 % (Fig. S52). No ZnO phases were detected in the XRD or PDF analysis, suggesting that minor changes in lattice parameters and electron density at the Bi site may be due to the introduction of Zn in the structure of $m$-BiVO₄ (potentially replacing Bi but also as interstitial dopants).

$^{51}V$ MAS NMR was collected for the ex-situ heated samples of **2**. The peaks of the molecular precursor broaden at 200 °C and largely disappear at 300 °C where $V_2O_5$ becomes the dominant species visible in the spectrum. A new $^{51}V$ NMR signal at −535 ppm becomes apparent after heating to 300 °C (Fig. 5a), and since these materials showed the presence of significant quantities of β-BiVO₄ in the 300 °C–400 °C range in the PDF data, we attribute this new species to β-BiVO₄. This signal is largely lost by 420 °C at which point only $V_2O_5$ (-615 ppm) and $m$-BiVO₄ (-425 ppm) remain, with both signals sharpening at higher temperatures ($V_2O_5$: FWHM of 17 and 11 ppm at 400 °C and 550 °C, respectively; $m$-BiVO₄: FWHM of 28 and 9 ppm at 400 °C and 550 °C, respectively), confirming that $m$-BiVO₄ becomes more crystalline at elevated temperatures. $^{51}V$ MAS NMR of annealed **3-Zn** shows the formation of $V_2O_5$ from 300°C along with a signal at −428 ppm, which we attribute $m$-BiVO₄ (Fig. 5b). However, in this case the $m$-BiVO₄ peak progressively disappears as the heating temperature increases, becoming invisible after heating at 600 °C. The associated PXRD data confirms the material contains $m$-BiVO₄, and this apparent discrepancy is due to the substantial broadening of the $^{51}V$ signal of $m$-BiVO₄ which occurs upon doping with $Zn^{2+}$, as previously reported in Sr-doped BiVO₄[76]. The aliovalent substitution of $Zn^{2+}$ for $Bi^{3+}$, possibly with added interstitial $Zn^{2+}$ sites for charge balance[15], induces substantial static disorder of the V sites causing the $^{51}V$ signal to broaden beyond detection under these measurement conditions[47,76].

In summary, **2** and **3-Zn** thermally transform through amorphous mixed-valence phases before the formation of small crystallites of BiVO₄ and $V_2O_5$ (Fig. 8). Initially two phases of BiVO₄ are identified at a stage when DMSO ligands have decomposed but S and some V(IV) are retained in the structure. On further heating the kinetic β-BiVO₄ phase is lost and the $ts$-BiVO₄ phase is retained (forming $m$-BiVO₄ on cooling).

Whilst no ammonia is present in these compounds, the presence of organic DMSO molecules can act as an internal reducing agent upon heating. Compared to decomposition of **1**, darker colours are retained during heating **2** to higher temperatures, suggesting the mixed valence is retained at later heating stages in this case. Volatile vanadium species are lost on annealing to increase the final BiVO₄:$V_2O_5$ ratios. MAS NMR data from **3-Zn** is strongly supportive of increasing amounts of Zn-doping into the resultant BiVO₄ phase with increasing temperature.

## Electrochemical properties of amorphous 1²⁰⁰ °C

Vanadium oxides have been extensively researched as Li-ion battery electrodes due to their ability to reversibly accommodate Li ions in their structure[50,77–80]. Particularly, $V_2O_5$ can reversibly intercalate up to two Li ions per unit formula due to its layered structure. However, insertion of an additional Li ion drives an irreversible phase change to a disordered rock salt structure, $\omega$-$Li_3V_2O_5$, which maintains good specific capacity, and can itself be used as a negative electrode in fast-charging Li- ion batteries due to its ability to intercalate additional Li ions[81]. Other vanadium oxides, like $\zeta$-$V_2O_5$ have been reported to deliver high discharge capacities of 250 mAh g⁻¹ in the voltage window between 4.0 and 2.0 V[82], whereas vanadyl phosphates like $\varepsilon$-VOPO₄ can deliver capacities exceeding 300 mAh g⁻¹ between 4.5 and 1.6 V due to the ability of vanadium to carry out multi-electron redox processes[83]. Motivated by these properties, a sample of black, mixed-valence **1²⁰⁰ °C**, which is easily prepared on the gram scale, was explored for its electrochemical properties as a Li-ion battery cathode. Initial slow cycling of **1²⁰⁰ °C** at 20 mA·g⁻¹ in the voltage window between 2.0 and 4.3 V showed a modest first cycle capacity of 150 mA·h·g⁻¹ which stabilised around 120 mA·h·g⁻¹ during the first few cycles (Fig. S53), with a smooth slopy voltage profile during the discharge curve, characteristic of a solid solution mechanism in disordered structures. Lowering the lower voltage cut-off to 1.0 V, however, reveals the appearance of a second plateau due to the intercalation of additional Li ions. In this voltage window, the specific discharge capacity of **1²⁰⁰ °C** reaches ca. 300 mA·h·g⁻¹ during the first cycle and steadily increases to 400 mA·h·g⁻¹ during the next 15 cycles (Fig. S53). To further investigate the cyclability of **1²⁰⁰ °C** under these conditions, we carried out cycling stability tests at 100 mA·g⁻¹ between 1.0 – 4.3 V (Fig. S54). Our results show an increase in specific discharge capacity from 60 to 210 mA·h·g⁻¹ during the first 100 cycles, an increase of more than three times the initial capacity, after which a slower steady decrease in specific capacity is observed (note the lower capacity due to the faster cycling of the material). This gradual increase in capacity suggests that initially amorphous **1²⁰⁰ °C** undergoes a series of structural transformations upon consecutive cycles (Fig. S55), which allow the structure to accommodate extra Li ions up to a maximum of ca. 2.75 Li per unit formula at 100 mA·g⁻¹. These structural changes are evident in the differential capacity curves, in which both oxidation and reduction features gradually shift to higher potentials, accompanied by an increase in capacity (Fig. S53). This is consistent with progressive, irreversible structural changes upon cycling, as observed in other vanadium oxide electrodes[84,85]. Unlike $V_2O_5$ or $(NH_4)V_4O_{10}$ which can accommodate Li ions in between its layers, the suggested majority $V_4O_9$ structure indicates that **1²⁰⁰ °C** would need to rearrange to be able to accommodate such amount of Li ions, and capacity may be further promoted by the transformation of remaining decavanadate ions in **1²⁰⁰ °C**. A similar behaviour has previously been reported for $V_4O_9$ in aqueous Zn-ion batteries, where structure amorphization has been observed upon $Zn^{2+}$ intercalation alongside a capacity increase during the first 100 cycles[86]. Additionally, previous studies of $(NH_4)V_4O_{10}$ and $V_4O_9$ in Li-ion battery cathodes have shown significant structural changes upon lithium intercalation with $V_4O_9$ irreversibly transforming into a disordered rock salt structure[59,62,87,88]. The observed specific capacities of **1²⁰⁰ °C** are in accordance with the reported $V_4O_9$, which can intercalate up to ~6 Li⁺ ( ~ 3.7-4 Li⁺ when cycled between 1.8 and 3.5 V)[62],

and with other reported mixed-valence vanadium oxides nanomaterials, which can incorporate multiple Li per formula unit and whose structure has been shown to change upon electrochemical cycling[85]. These possible structural transformations are also reflected in the evolution observed in the voltage profile of $1^{200\,°C}$, which shows the appearance of two distinct steps in the charge curve, suggesting that at least two structural processes take place upon Li deintercalation. Even though we cannot discard the contribution of small amounts of amorphous POVs in, these results show that amorphous vanadium oxides can display intriguing electrochemical properties as Li-ion battery electrodes allowing for the reversible intercalation of an increasing amount of Li ions.

In conclusion, we have shown that thermal decomposition of mono-, bi-, and trimetallic POV molecular precursors follows complex pathways involving amorphous and intermediate phases, as characterised by a wide range of analytical methods including solid-state NMR, X-ray PDF analysis, and variable temperature X-ray diffraction. This approach enabled us to identify previously unknown mixed-valence $V_4O_9$-like amorphous intermediates exhibiting strong colouration and paramagnetism. Our results show the potential of using temperature control of the decomposition process to achieve specific vanadium oxidation states, relating to oxygen vacancy content in the final materials[89,90], and to tune the degree of crystallinity and particle size.

When forming $BiVO_4$ materials, the V-rich SSPs used mitigate vanadium loss during heating, ensuring no Bi-rich oxide byproducts are formed. Furthermore, solid-state NMR studies reveal temperature-dependent incorporation of Zn dopants into the structure of $BiVO_4$ when a Zn containing trimetallic SSP is chosen, with associated site disorder found in the doped structure. This establishes a direct link between thermal processing and atomic-scale structural evolution.

Finally, we have discovered a kinetically stabilised polymorph of $BiVO_4$, dubbed $β$-$BiVO_4$, which is structurally analogous to $β$-$SnWO_4$ and accessible exclusively via controlled thermal decomposition of SSPs (and remains stable after cooling). Interestingly, the coordination geometry of Bi in $β$-$BiVO_4$ resembles that in the molecular precursor more closely than other $BiVO_4$ phases. Theoretical analysis suggests the $β$-$BiVO_4$ phase displays an increased band-gap relative to $m$-$BiVO_4$, in part due to a less dense structure which weakens V 3 d orbital overlap and narrows the conduction band, whilst reduced involvement of the Bi 6 s in bonding, reduces the energy of the highest energy valence band energy states. This highlights the fascinatingly different properties in polymorphs of materials and our results show the potential of POV molecular precursors for accessing novel oxide phases with tailored properties.

More widely, this study showcases the possibilities of uncovering a multitude of material phases during molecule-to-material transformations. The use of specific and tuneable precursors which transform to materials at low temperatures will surely allow access to further metastable phases yet to be discovered experimentally and suggests many further exciting discoveries are possible in this space.

## Methods
### Preparation of precursors
Ammonium metavanadate, $[NH_4][VO_3]$, (Sigma-Aldrich, >99%) was used directly from suppliers. $[VO(O^nPr)_3]$ (Sigma-Aldrich, 98%), $Bi(NO_3)_2·5H_2O$ (Sigma-Aldrich, 98%). and $ZnCl_2$ (Sigma-Aldrich) were stored in a $N_2$-filled glovebox. Dimethyl sulfoxide (DMSO) was degassed before use and stored over 4 Å molecular sieves. All other solvents were used without further purification. All reactions using a Schlenk tube were performed under inert conditions ($N_2$ atmosphere).

Compound **1** was synthesised based on established methods originally proposed by Lacharte et al[91], full details can be found in the supporting information. Elemental analysis (predicted): % H, 3.03 (3.09); % N, 6.64 (7.16).

**2** and **3-Zn** were prepared following literature procedures[15], full details can be found in the supporting information. For **2**, elemental analysis (predicted): % C, 11.07 (11.33); % H, 2.49 (2.94); % S 15.14 (15.13), ICP-OES Bi: V metal ratio (predicted): 13: 3.6 (13: 4). For **3-Zn**, elemental analysis (predicted, $[Bi_2V_{12}O_{33}Cl(DMSO)_6]_2$ $[Zn(DMSO)_6]·12(DMSO)$): % C, 10.50 (12.88); % H, 2.54 (3.24); % S 14.52 (17.19). Note that sample has less co-crystallised DMSO that previously reported, and C, H, S analysis of this sample is a better match for $[Bi_2V_{12}O_{33}Cl(DMSO)_6]_2[Zn(DMSO)_6]·4(DMSO)$ (calc. %: C, 10.63; H, 2.68; S, 14.19)[15]. N.B. Grinding the microcrystalline sample and placing it under vacuum overnight led to an amorphous sample which had slightly less co-crystallised DMSO (Elemental analysis, % C, 10.41; H, 2.54; S, 12.97). ICP-OES Bi: V: Zn metal ratio (predicted): 24: 3.9: 0.9 (24: 4: 1).

$[VO(DMSO)_5][NO_3]_2$ and **3-VO** were prepared as crystalline materials; synthetic details, crystal structure information and futher analysis is available in Supporting Note 1 in the supporting information.

### Annealing of precursors to extended materials
**1 to $V_2O_5$. 1** (~0.25 g) was added to an alumina crucible and placed in a furnace. It was heated to varying temperatures using a heating ramp rate of 10 °C min⁻¹. Each sample was held at the specified final temperature for two hours.

**2 or 3-Zn to $BiVO_4$ / $V_2O_5$. 2** or **3-Zn** (~0.25 g) was added to an alumina crucible and placed in a furnace. It was heated to varying temperatures using a heating ramp rate of 10 °C min⁻¹. Each sample was initially held at 175 °C for two hours to decompose most of the associated DMSO, then the ramping rate was continued until the desired final temperature was reached and this temperature was then held for an additional two hours.

**Powder X-ray diffraction (PXRD).** PXRD patterns of the materials were recorded using a Panalytical Empyrean diffractometer with a Cu target ($K_{α1}$ = 1.54056 Å) using background-free holders made of monocrystalline silicon. Variable temperature PXRD was performed on a Panalytical X'Pert Pro MPD equipped with a curved Ge Johansson monochromator, giving pure Cu $K_{α1}$ radiation, and a solid state PiXcel detector running in 1D scanning mode. The sample was mounted in an Anton Paar HTK1200N furnace and aligned to the half-cut direct beam at room temperature to ensure it was in the centre of rotation of the goniometer. The height was then automatically adjusted using the Anton Paar software to account for the thermal expansion of the stage with temperature. The samples were heated at a rate of 5 °C/min with a wait time of 10 min before each measurement. Scans were made between 5°–70° 2θ with a step size of ~0.0131° and a count time of ~300 s per step. The lower energy threshold of the detector was adjusted to avoid measuring the fluorescence that comes from V under Cu radiation. Rietveld refinements were carried out using the software Topas Academic v7 (http://www.topas-academic.net/)[92] and the $V_2O_5$ (ICSD 60767), $m$-$BiVO_4$ (ICSD 100162), $t$-$BiVO_4$ (ICSD 144629) phases and instrumental parameters were obtained from the Rietveld refinement of a Si standard[92].

**X-ray pair distribution function (XPDF) measurements.** Total scattering data were collected at the I15-1 XPDF Beamline at Diamond Light Source, U.K. Powder samples were loaded into borosilicate glass capillaries (1.0 mm) and spun perpendicularly to the beam to improve powder averaging during data collection. Total scattering data collection was measured with a PerkinElmer area detector at an X-ray energy of 76.69 keV and data collection time of 600 s. The data was processed, and Fourier transformed using PDFGetX3 in the $0.5 < Q < 22.0$ Å⁻¹ range to obtain the final PDF as $G(r)$[93]. PDF refinements were carried out using the PDFGui software[94] using the $V_2O_5$ (ICSD 60767), $m$-$BiVO_4$ (ICSD 100162), $V_2O_9$ (ICSD 15041), $(NH_4)V_4O_{10}$ (ICSD 230493) and $(NH_4)_6V_{10}O_{28}$ (ICSD 161217).

**Solid-state NMR spectroscopy.** Solid-state MAS NMR spectra of $^1$H (500.1 MHz) and $^{51}$V (131.5 MHz) were recorded at room temperature on a Bruker Avance III 11.7 T spectrometer equipped with 2.5 and 3.2 mm MAS probes. $^1$H spectra were recorded using an echo sequence. $^{51}$V spectra were recorded using single pulse excitation with a short (1 μs) pulse at 80-100 W RF power. $^{13}$C (213.8 MHz) spectra were recorded at room temperature on a Bruker Avance Neo 20 T spectrometer equipped with 3.2 mm MAS probe. $^{51}$V shifts were referenced using the unified IUPAC recommendation using solid adamantane as a secondary reference (38.48 ppm for the $^{13}$C of the $CH_2$ carbon). All acquisition and processing details are given in Table S1.

**Electrochemical measurements.** $1^{200\ °C}$ coatings for electrochemical cell testing were prepared by slurry casting onto an Al foil. The slurry was obtained by mixing 80 wt.% of active material (AM), 10 wt.% of Super C65 Carbon black, and 10 wt.% of polyvinylidene fluoride (PVDF, Merck) as binder with N-methyl-2-pyrrolidone. The resulting coating was dried at 110 °C overnight in a vacuum oven before use. Electrochemical tests were performed by preparing CR2032-type coin half cells using the coated cathode (14 mm diameter, mass loading: 2-3 mg cm$^2$), Li metal anode (15 mm, Merck), Whatman glass microfiber separator (18 mm) and 1.0 M LiPF$_6$ in EC:EMC (ethylene carbonate/methyl carbonate, 3:7 V/V), which were assembled inside an Ar-filled glovebox. Galvanostatic cycling tests were performed in the voltage ranges of 2.0–4.3 V and 1.0 and 4.3 V at current densities of 20 mA·g$^{-1}$ and 100 mA·g$^{-1}$.

**Solid-state UV-vis spectroscopy.** UV-Vis spectra were recorded using Shimadzu 2600i spectrometer in the diffuse reflectance configuration.

**FT-IR spectroscopy.** IR spectra were collected using a Agilent Technologies Cary 630 FT-IR spectrometer in the solid-state using an attenuated total reflectance setup.

**C, H, N, S elemental analysis.** Samples were sent for analysis at London Metropolitan University.

**Inductively coupled plasma – mass spectrometry (ICP-MS).** Samples were dissolved in HNO$_3$ (precursors) or HCl (annealed oxides) and diluted with water for V:Bi:Zn ratio analysis using an Agilent 7900 ICP-MS.

**Thermogravimetric analysis – mass spectrometry (TGA-MS).** Samples were heated from room temperature to 800 °C or 1000 °C with a ramping rate of 5 °C or 10 °C min$^{-1}$ using a Mettler Toledo TGA/DSC 1 instrument under N$_2$ or air respectively. Samples were heated in the presence of N$_2$ or air with a flow rate of 20 mL min$^{-1}$.

**EPR spectroscopy.** X-band ( ~ 9.5 GHz) spectra were recorded on a Bruker EMX spectrometer at room temperature.

**XANES.** The V K-edge X-ray absorption near edge structure (XANES) data of the powder samples were collected on an easyXAFS300+ spectrometer. All measurements were performed in transmission using a silver X-ray source operated at 40 kV and 10 mA. Air scattering was minimised using a helium chamber.

## Data availability
Supplementary information (XRD, details of NMR experiments, additional NMR spectra, computational details and additional DFT results) is available in the online version of the paper. The source NMR data are available on Zenodo: https://zenodo.org/records/19102456 and XRD source data are available on the Warwick Research Archive Portal (https://wrap.warwick.ac.uk/194481/). Crystallographic data for the structures reported in this Article have been deposited at the Cambridge Crystallographic Data Centre, under deposition numbers CCDC 2391125 ([VO(DMSO)$_5$][NO$_3$]$_2$) & 2391126 (**3-VO**). Copies of the data can be obtained free of charge via https://www.ccdc.cam.ac.uk/structures/.'

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

## Acknowledgements

S.D.P. and A.E.H. are grateful for financial support from the Royal Society through a Royal Society University Research Fellowship (URF\R1\191458) and Research Grant (RG\R2\232264). Thanks to the University of Warwick CDT in Analytical Science and the EPSRC Doctoral Training Partnership grant for studentships for T.J.B. and S.E.B. respectively. D.J.K. acknowledges the UKRI Horizon Europe guarantee funding (PhotoPeroNMR, grant agreement number EP/Y01376X/1). The UK High-Field Solid-State NMR Facility used in this research was funded by EPSRC and BBSRC (EP/T015063/1) as well as the University of Warwick including via part funding through Birmingham Science City Advanced Materials Projects 1 and 2 supported by Advantage West Midlands (AWM) and the European Regional Development Fund (ERDF). J.C-G. thanks The Leverhulme Trust for an Early Career Fellowship (ECF-2023–129). J.W. acknowledges funding from the Knut and Alice Wallenberg Foundation (Nos.~2023.0032 and 2024.0042), the Swedish Strategic Research Foundation (FFL21-0129), and the European Research Council (ERC Starting Grant No. 101162195). Computational resources were provided by the National Academic Infrastructure for Supercomputing in Sweden at NSC, PDC, and C3SE. X-ray diffraction measurements were made at University of Warwick X-ray Diffraction Research Technology Platform, which is part of the Warwick Analytical Science Centre supported by EPSRC (EP/V007688/1). We thank Dr James Town, Dr Lijiang Song for recording TGA-MS, ICP-MS data respectively and Eniola Sokalu for assistance with experiments. We acknowledge Diamond Light Source for providing beamtime on I15–1 (Experiments CY26330-8, CY26330-9).

## Author contributions

Original concept by S.D.P. Synthesis and characterisation of samples by A.E.H., T.J.B., E.V., and S.D.P. Solid-state NMR spectroscopy by A.S., B.M.G., D.J.K. EPR spectroscopy by S.E.B. DFT calculations by J.W. Variable temperature PXRD experiments by A.E.H. and D.W. XANES by A.S.M. PXRD and

PDF analysis and fitting by J.C.G. Single crystal X-ray crystallography by T.J.B. and S.E.B. S.D.P., J.C.G., D.J.K., and J.W. wrote the article.

## Competing interests

The authors declare no competing interests.
