## [Transparent Peer Review file · Nature Communications]

Amorphous intermediates and discovery of a kinetic polymorph of BiVO₄ from heating V+Bi+Zn single-source precursors

Corresponding Author: Dr Sebastian Pike

Version 0:

Reviewer comments:

Reviewer #1

(Remarks to the Author)

Manuscript no: NCOMMS-25-61777

Title: Thermal transformation of V+Bi+Zn single-source precursors: identification of amorphous intermediates and a new kinetic phase of BiVO₄.

Thank you for the opportunity to review this manuscript. The work presents an interesting and well-structured study on the thermal decomposition of mono-, bi-, and trimetallic polyoxovanadates (POVs), offering valuable insights into intermediate species, polymorph formation, and electrochemical properties. The results, particularly the identification of a previously unknown β -BiVO₄ polymorph and the discussion of Zn dopant incorporation, are noteworthy and contribute to advancing the understanding of molecular single-source precursor strategies. However, I have several comments and requests for clarification that I believe will improve the clarity, accuracy, and reproducibility of the manuscript. My specific comments are as follows:

1. At line 39, the authors state that BiVO₄ is an important photocathode material with optimal band-gap properties for water oxidation. However, this appears to be a typographical or conceptual error, as BiVO₄ is widely recognized and extensively reported in the literature as a photoanode material for water oxidation.
2. At line 113, the authors state that a mass loss of 22.5% is expected upon formation of V₂O₅ during thermal decomposition. However, the manuscript does not explain how this specific value (22.5%) was calculated or derived.
3. At line 128, the authors describe a progressive color change of the samples, from orange to brown (after 150 °C), to black (after 200 °C), and then to red/brown after 300 °C (Fig. S11). However, Fig. S11 currently shows only the images at 200 °C and 400 °C. Since a notable color change is reported to occur at 300 °C, it is recommended that the authors include the corresponding image at 300 °C in the supplementary figure for clarity and completeness.
4. In Figure S18, the Powder XRD pattern of compound 2 is compared with a simulated pattern derived from the single-crystal structure. However, the manuscript does not specify the software, database, or methodology used to generate the simulated pattern. It is recommended that the authors provide these details, including any parameters or settings applied, to ensure reproducibility and clarity for readers.
5. In Figure 5a, for the MAS NMR spectra of compound 2 at 300 °C, two arrows are indicated in the figure. However, the manuscript does not clarify what these arrows are intended to highlight. Could the authors please elaborate on the specific features or observations that these arrows are meant to emphasize for better clarity and interpretation of the data?
6. In the variable-temperature analysis, such as PXRD experiments of compound 1 (25–180 °C) in Figure S6, we have some concerns regarding temperature accuracy. Furnaces typically have larger fluctuations at lower temperatures (<200 °C), which may affect data reliability. Could the authors clarify how temperature stability was monitored or controlled during these measurements?
7. While the new kinetic phase β -BiVO₄ is convincingly described, the authors did not explore its electronic/photophysical properties. Could the authors provide some preliminary data or discussion (e.g., optical bandgap, DFT predictions, photocurrent benchmarking) to support claims of possible photocatalytic interest? Secondly, how reproducible is the formation of β -BiVO₄ across batches and heating conditions? Can it be stabilised or isolated at larger scale?
8. V₄O₉-like structure was assigned to the black 200 °C phase of ammonium decavanadate. However, given the low detected V(IV) content (~3%), could alternative explanations (e.g., disordered V₂O₅ with defects) also fit the data? Besides, the colouration of the intermediate is attributed to IVCT. Could this be proven in the UV-Vis absorption spectra?

9. In the electrochemical testing, cycling shows increasing capacity up to 400 mAh g⁻¹. Could the authors clarify the mechanism (structural transformation vs activation)? Are there ex-situ characterisation data after cycling to support this? How reproducible are the cycling data across samples?

10. The manuscript frames BiVO₄ largely as a photoanode material, but the new β-BiVO₄ phase is only structurally described. Could the authors better contextualise how this discovery fits into the broader landscape of polymorph engineering for photocatalysis? For example, can this be extended to stabilising other metastable polymorphs via SSP routes? Similarly, how does this work compare with other recent reports on metastable BiVO₄ phases?

Reviewer #2

(Remarks to the Author)

The authors report very interesting results of thermal decomposition reactions using mono-, bi- and trimetallic polyoxovanadates (POVs). Upon heating, all three compounds lose volatile species and amorphous intermediates are formed. Increasing the temperature crystalline V₂O₅ and BiVO₄ developed and a new kinetically stabilized polymorph of BiVO₄ was identified, which is named β-BiVO₄. A variety of methods were applied to shed light on local structures of the intermediate amorphous phases. Final crystalline materials were characterized with Rietveld refinements of XRD data. State-of-the-art methods include PDF analysis, in-situ X-ray scattering, solid state NMR, EPR, chemical analyses, UV-Vis, FT-IR spectroscopy, XANES, TGA-MS, and DSC. At the end, the authors were able to develop a comprehensive picture about the amorphous phases and to identify the new polymorph of BiVO₄. The investigations, experiments and interpretation of the data go far beyond what the reviewer has seen in literature. Before the very well written manuscript can be accepted a revision is required.

First a personal remark of the reviewer: the part dealing with the results of Li intercalation/insertion into 1-200 is out of place and does not add information about the main aim of the manuscript: understand phase formation during thermal decomposition, identify local and long-range structures related to chemical compositions, identify new crystalline compounds. But it is the decision of the authors whether they still want to include this part in a revised version. In this case much more information about the electrochemical experiments must be supplied.

l. 50: the thermal decomposition of ammonium decavanadate hexahydrate was investigated by Todorović, Mioč, Holclajtner-Antunović and D. Šegan; please add the reference

l. 85: please add the chemical formula of compound 2.

l. 100/101: if the sample is a 1:1 crystalline phase mixture one would expect that the PXRD patterns shows reflections of both components. Fig. S2 shows only the calculated pattern of one of the two components. A comment on this is required. If possible please rescale the calculated pattern.

l. 111: ... Thermogravimetric analysis (TGA-MS/DSC) shows several consecutive mass...In Fig. S3 the DTG curves should be included helping to identify the distinct mass loss steps. Fig. S4: please include an arrow showing the endo-exo direction. At least in the SI a short discussion of what is seen in the DSC curves should be included. What is the sharp event occurring just below 700 °C in the DSC curve recorded in air? Assignment of emitted masses: is really atomic oxygen emitted or is it NH₂? Fig. S5: why were different heating rates chosen for the experiments in air and N₂? Fig. S7: please add a Table summarizing results of the Rietveld refinements.

l. 118-121: the reviewer suggests performing the analyses on a sample obtained at identical temperature. Here XRPD was measured on a samples obtained at 160 °C, IR on a sample heated to 175 °C and H, N analysis was done on a sample heated to 150 °C. This is somewhat disturbing.

l. 125: ... powdered 1 was estimated to have average crystallite sizes of ~10 nm...at what temperature?

l. 129: is it not possible estimating the bandgaps from the UV-Vis data (Kubelka-Munk or tauc plots)? (see also Fig. S27). Why are no UV-Vis data are provided for 3-Zn and the decomposition products?

l. 130:... ruling out carbon contamination...where should the carbon come from? 1 contains no carbon.

l. 139: The presence of V(IV) was confirmed by EPR (Fig S13). Is it not possible estimating the amount V(IV) from the spectrum?

l. 217: ... noteworthy that precipitation of 2 may generate an amorphous form (2amor) which...Here (or in the SI) the procedure for obtaining 2amor should be described (the reviewer has not found information about this)

p. 10: the authors recoded NMR spectra of 2, on p. 11 TGA was performed on 2amor and the authors write that 520 °C is required to complete the mass loss for 2. This is a little bit confusing. Please clarify.

l. 231: ... decomposition (and water is also lost from 2, Fig S18-26)...Fig. S18 displays the PXRD pattern of 2

l. 236: ... indication of CO₂, H₂CO, SO₂ and H₂S loss occurring...In Fig. 23 no mass is indicated for H₂S

The DSC curves (Fig. S22) should be discussed at least in the SI because significant differences are seen in the traces for air and N₂.

l. 266, Fig. 6: please adjust the chemical formula and the cubic space group

l. 292: it is a lone electron pair

l. 322/323: one digit is enough for the estimated uncertainty/standard deviation

l. 326: Table S8

l. 338: 300 °C instead of deg C

l. 340-351: please indicate what polymorph of BiVO₄ is meant

l. 354/355: beta-BiVO₄ instead of b-BiVO₄

Fig. S19: there are remarkable intensity differences between experimental and calculated XRD patterns for 3-Zn. A comment is required.

Fig. S4 and S22: include an arrow showing the endo-exo direction

Fig. S23: the assignment of the masses should be reconsidered. 2amor does not contain nitrogen. See also Fig. S26; what is the dip in the mass traces at around 300 °C?

Fig. S24: please show also the DTG curves.

Fig. S44: It seems that the figure caption contains duplicate information.

In the Method section the authors should mention that crystals with composition $[\text{VO}(\text{DMSO})_5][\text{NO}_3]_2$ and 3-VO were obtained and the crystal structures were determined, more information is given in the see SI.

Reviewer #3

(Remarks to the Author)

The authors report on different single source precursors containing V, V+Bi or V+Bi+Zn, which they thermally transform to intermediate phases and final oxides. They follow the transformation by ex situ and in situ characterization methods, which also enable the study of the atomic structures of amorphous and disordered phases. The aim of the work was to obtain multi-element solids directly from the precursor. In case it was the goal of the studies to synthesize one-structure multi-metal oxides, it failed e.g. as the 3-Zn-V-Bi system formed phase mixtures. The authors also tested the electrochemical properties of the vanadium system, but the results are not discussed in relation to literature reports on the same composition (e.g., DOI <https://doi.org/10.1039/D1QI00747E>).

For compounds 2 and 3-Zn, the manuscript provides some insights into the crystallization pathways, but the structure-property relationship was not investigated. The chapters on sample 1 and on sample 2/3-Zn are therefore not really related. The authors claim to be able to adjust the oxidation state of vanadium through synthesis. In my opinion, however, this does not justify publication in Nature Communications. For these reasons, I recommend submission to a more specialized journal. General comments:

1. The authors apply a variety of techniques, which sometimes makes the manuscript difficult to follow. The manuscript could be more structured to allow identification of the differences and similarities between the different systems.
2. The quality of the experimental data varies significantly. For example, Figure S19 plots all XRD patterns for different 2θ ranges.
3. The authors also provide a large number of figures in the supplementary information (SI). Nevertheless, they should be more consistent in terms of representation. For example, some figures use '2 theta /(degrees)' and others use "Two theta ($^\circ$) or 2θ / $^\circ$ ".
4. The figures have different fonts used for labelling the axes, etc. This makes the manuscript appear inconsistent. It looks as if different authors have contributed different measurements and everything has simply been put together in one file.
5. XRPD in Figures 2 and 6a: The details are difficult to discern. The details that could indicate intermediate phases are not visible. Such figures are not helpful at all. I recommend adding the Rietveld refinement plots as individual plots to the Supporting Information, scaled reasonably to allow the difference curves and simulations to be seen. The pattern should appear as shown in Figure S48.
6. The discussion of the ex situ and in situ XRD patterns is very difficult to follow, especially for compounds 2 and 3Zn. One has to jump back and forth between the main manuscript and the SI.
7. For clarity, I would plot the XRDs of the starting materials together with the calcined samples.
8. Are the ex situ and in situ phases the same for the given temperatures? Do reversible HT phases transform back to LT phases during cooling? I think this will be addressed later but it would be helpful to reconsider the structure of the discussion.
9. It is also difficult to follow the phase transformation pathways when samples 2 and 3-Zn are discussed in the same sentence.
10. Explanation of 2 or 3-M is missing
11. Crystal structure data used for XRD and PDF refinements (ICSD numbers etc)
12. A PDF of the starting compound would be helpful for following the transformations.
13. PDF: Page 6, line 134: "1-200 $^\circ\text{C}$ reveals a different PDF pattern, which was found to resemble V4O9 (Pnma,53, 54 Rw = 30.4% after refining thermal factors and atomic positions, Fig S12)."
Comment: please provide a table showing the refined Usio values, lattice parameters, and refined atomic positions, including the resulting main interatomic distances.
14. PDF: Page 6, line 135: "The fit was improved by adding a contribution from either decavanadate or hexavanadate anions (Rw = 20.1%, (1) = 21.4 wt%, Fig 3b; or Rw = 26.7%, ($[\text{NH}_4]_2[\text{V}_6\text{O}_{16}]$) = 13.2 wt% respectively),..."
Comment: Which structure was included? To my opinion, the refinement plots in Figures 3b and S12 do not differ significantly.
15. Page 6, line 139: "The presence of V(IV) was confirmed by EPR (Fig S13). 1-200 $^\circ\text{C}$ was dissolved in 2M H₂SO₄ and a double titration method used to identify V(IV) content,55 which indicated only 2.7 ± 1 % of V was V(IV). V₂O₅, 1 and 1-200 $^\circ\text{C}$ also appear almost identical by X-ray Absorption Near Edge Structure (XANES) analysis, consistent with a majority of V(V) in 1-200 $^\circ\text{C}$ (Fig S14)."
Comment: EPR confirms the presence of V₄₊, but XAS shows that mainly V₅₊ is present.
16. How do these results compare with those of the PDF, which shows a mixture of V₄₊ and V₅₊ for V₄O₉? The authors discuss this later, proposing that V₄₊ be replaced with V₅₊. However, perhaps this is too simple an explanation.
17. Page 7, line 153: "In 1-200 $^\circ\text{C}$ the local connectivity resembles V₄O₉ but the low V(IV) content implies that most sites are occupied by V(V), implying that this may be a kinetically stabilized phase. In related solution studies, heating $[\text{NMe}_4][\text{H}_2\text{V}_{10}\text{O}_{28}]$ under hydrothermal conditions is reported to form mixed valence $[\text{NMe}_4][\text{V}(\text{V})_3\text{V}(\text{IV})\text{O}_{10}]$ with a layered structure.^{50, 57}"
Comment: Why have the authors not simulated this structure?
18. Electrochemical measurements suggest the presence of other phases in the sample calcined at 200 $^\circ\text{C}$, such as V₆O₁₃, but confirmation of these phases is missing.
19. The thermally induced structure transformations for sample 1 are not convincingly supported by experimental PDF data, which show only V₄O₉.
20. Page 10, line 216: "...including the newly discovered structure $[\text{Bi}_2\text{V}_{12}\text{O}_{33}\text{Cl}(\text{DMSO})_6]_2[\text{VO}(\text{DMSO})_5]_8(\text{DMSO})_3 \cdot 3\text{-VO}$."

Comment: Reference for the crystal structure is missing.

21. Page 14, line 286: " This gave excellent results, confirming that this new kinetic phase crystallizes in the cubic space group P213 with cell parameters $a = 6.9916(2)$ Å with Bi and V atoms sitting in the 3-fold symmetry axis of the cubic unit cell. Therefore, we dub this new phase β -BiVO₄."

Comment: it should be "cell parameter" instead of "cell parameters", as only one unit cell parameter is required for a cubic space group.

22. An atom cannot "sit" "in" an axis, but it can be located on a symmetry element.

23. Page 14line 309: "2amor primarily yields tetragonal BiVO₄ (space group I41/a) at 300 °C followed by the subsequent crystallisation of V₂O₅ as the temperature increases (Fig S32), in contrast, for microcrystalline 3-Zn, V₂O₅ appears as the major species at 300°C before BiVO₄ phases grow in (Fig 6b)."

Comment: 2amor primarily yields tetragonal BiVO₄ (space group I41/a) at 300 °C followed by the subsequent crystallisation of V₂O₅ as the temperature increases (Fig S32). In contrast, for microcrystalline 3-Zn, V₂O₅ appears as the major species at 300°C before BiVO₄ crystallises (Fig 6b)."

24. Page 15, line 322: "For 2 after 400°C, PDF analysis suggests 42.8(55) wt% is β -BiVO₄, however this drops significantly by 420°C, at which point weight fractions are 43.4(37)% V₂O₅ : 52.6(35)% BiVO₄ : 4.0(18)% β -BiVO₄."

Comment: If the estimated standard deviation is larger than the refined main value (43.4(37%)), this should raise suspicions. Or was the standard deviation not used correctly (43.4(4))?

25. For sample 1, electrochemical data were discussed, for samples 2 and 3-Zn, no data are obviously available. The entire discussion focuses on structure transformations. Again, this appears inconsistent with the first part.

26. The graphical presentation of the thermal pathways is somewhat puzzling. Usually, one would go from the bottom to the top, from blue to red, rather than downhill from the small crystals to the oxides.

Version 1:

Reviewer comments:

Reviewer #2

(Remarks to the Author)

The revised version of the manuscript is significantly improved adding new information and data, especially results of theoretical calculations. The authors carefully considered my suggestions and critics, and the reviewer is fully satisfied with the present version. Hence, acceptance of the paper is highly recommended.

The authors use the abbreviation Fig with full stop or without. Please homogenize.

Reviewer #3

(Remarks to the Author)

The authors revised the manuscript were carefully and they included numerous additional information and discussions. Based on the changes I consider the manuscript suitable for publication.

RESPONSE TO REVIEWERS' COMMENTS

We thank all the reviewers for their time and careful consideration of the manuscript. We are grateful for the many excellent suggestions which have significantly strengthened the work.

Reviewer #1 (Remarks to the Author):

Manuscript no: NCOMMS-25-61777

Title: Thermal transformation of V+Bi+Zn single-source precursors: identification of amorphous intermediates and a new kinetic phase of BiVO₄.

Thank you for the opportunity to review this manuscript. The work presents an interesting and well-structured study on the thermal decomposition of mono-, bi-, and trimetallic polyoxovanadates (POVs), offering valuable insights into intermediate species, polymorph formation, and electrochemical properties. The results, particularly the identification of a previously unknown β -BiVO₄ polymorph and the discussion of Zn dopant incorporation, are noteworthy and contribute to advancing the understanding of molecular single-source precursor strategies. However, I have several comments and requests for clarification that I believe will improve the clarity, accuracy, and reproducibility of the manuscript. My specific comments are as follows:

Thank you for the very supportive comments, we appreciate that the reviewer believes the paper is well structured and will contribute to the field.

1. At line 39, the authors state that BiVO₄ is an important photocathode material with optimal band-gap properties for water oxidation. However, this appears to be a typographical or conceptual error, as BiVO₄ is widely recognized and extensively reported in the literature as a photoanode material for water oxidation.

Thank you very much for spotting this typographical error, the statement has been corrected.

2. At line 113, the authors state that a mass loss of 22.5% is expected upon formation of V_2O_5 during thermal decomposition. However, the manuscript does not explain how this specific value (22.5%) was calculated or derived.

This value is derived from the % mass of the $6(NH_4) + 3O + 6(H_2O)$ components in compound 1, which are lost as NH_3 and H_2O on annealing to leave only the V and O atoms as V_2O_5 . This is now clarified in the text.

3. At line 128, the authors describe a progressive color change of the samples, from orange to brown (after 150 °C), to black (after 200 °C), and then to red/brown after 300 °C (Fig. S11). However, Fig. S11 currently shows only the images at 200 °C and 400 °C. Since a notable color change is reported to occur at 300 °C, it is recommended that the authors include the corresponding image at 300 °C in the supplementary figure for clarity and completeness.

A photograph after heating to 300°C has been added to Fig S11

4. In Figure S18, the Powder XRD pattern of compound 2 is compared with a simulated pattern derived from the single-crystal structure. However, the manuscript does not specify the software, database, or methodology used to generate the simulated pattern. It is recommended that the authors provide these details, including any parameters or settings applied, to ensure reproducibility and clarity for readers.

Thank you for highlighting this. The simulated data was generated using Mercury, from a dataset originally published by Streb et al. This information is added to the caption of Fig S18.

5. In Figure 5a, for the MAS NMR spectra of compound 2 at 300 °C, two arrows are indicated in the figure. However, the manuscript does not clarify what these arrows are intended to highlight. Could the authors please elaborate on the specific features or observations that these arrows are meant to emphasize for better clarity and interpretation of the data?

We have revised the caption to figure 5 to clarify this point: "The signals attributed to β - $BiVO_4$ are indicated by arrows, which indicate the possible positions of the isotropic resonance and a spinning sideband. While β - $BiVO_4$ is expected to contain only one distinct V site, the large linewidth makes the identification of the isotropic peak challenging."

6. In the variable-temperature analysis, such as PXRD experiments of compound 1 (25–180 °C) in Figure S6, we have some concerns regarding temperature accuracy. Furnaces typically have larger fluctuations at lower temperatures (<200 °C), which may affect data

reliability. Could the authors clarify how temperature stability was monitored or controlled during these measurements?

Thank you for checking this important point. In the variable temperature PXRD studies the samples were mounted in an Anton Paar HTK1200N furnace. The samples were heated at a rate of 5° C/min with a wait time of 10 minutes before each measurement.

The furnace temperature is controlled to hold at the correct temperature with a stability of < 1° C. The temperature is also measured and controlled in this case by a thermocouple placed directly beneath where the sample is mounted.

Based on previous measurements, we know this furnace setup is highly reproducible, i.e. any errors are typically systematic rather than random (fluctuations). In previous it has been accurate to < +/-5C at the temperature range measured in this study.

7. While the new kinetic phase β -BiVO₄ is convincingly described, the authors did not explore its electronic/photophysical properties. Could the authors provide some preliminary data or discussion (e.g., optical bandgap, DFT predictions, photocurrent benchmarking) to support claims of possible photocatalytic interest?

To address this point, we now include a new co-author on the project bringing expertise of computational study of BiVO₄ materials. We have now carried out DFT calculations of the β -BiVO₄ polymorph and have found that the band gap is significantly larger than that of the monoclinic phase: 3.34 eV vs 2.27 eV within r2SCAN, and 4.78 eV vs 3.20 eV within PBE0(α). β -BiVO₄ has a less pronounced excitonic peak, but both materials have similar excitonic binding energies (about 0.4 eV). We have now included the predicted density of states of the new polymorph as Figure 7, as well as band structures and absorption spectrum in Figures S44-S46."

Secondly, how reproducible is the formation of β -BiVO₄ across batches and heating conditions? Can it be stabilised or isolated at larger scale?

Thank you for these excellent questions. The beta phase forms reproducibly across different batches and conditions (including from different precursors – 2, 3-Zn and 3-VO) but noting that we report a difference in the % composition of the beta phase when using amorphous 2 versus crystalline 2, therefore, we suggest that crystallinity of the precursors plays a role in the stabilisation of the kinetic phase. At present we are not able to isolate the beta phase at a larger scale.

8. V₄O₉-like structure was assigned to the black 200 °C phase of ammonium decavanadate. However, given the low detected V(IV) content (~3%), could alternative explanations (e.g., disordered V₂O₅ with defects) also fit the data?

We thank the referee for raising this point. Our first option was indeed to try the orthorhombic V₂O₅ phase identified by Rietveld and PDF refinement of 1 annealed at 400 °C. However, using

this model resulted in a very poor fit to the experimental PDF ($R_w = 80.4\%$), even after refining the partial site occupancy of V atoms to account for possible V vacancy defects. Additionally, refining the atomic positions resulted in non-physical interatomic distances. Below there is a representative fit using the orthorhombic V_2O_5 model (Fig R1).

Figure R1. Representative fit of 1-200 using the V_2O_5 phase ($R_w = 87.1\%$, $Pnmm$, ICSD 60767)

To comply with the reviewer's comment, we also tested the other V_2O_5 phases available in the ICSD and evaluated the fits in terms of the goodness of fit (R_w), and 'stability' of the model, i.e. checking whether any of the structural parameters became non-physical during the refinement. However, none of them provided satisfactory fits. **Table R1** gives a summary of all the V_2O_5 phases tested and the value of R_w agreement factor.

Table R1 – PDF fit agreement factors from different V_2O_5 models against 1-200 experimental PDF data.

ICSD Code	Space group	$R_w / \%$
60767	$Pmnm$	87.1
44212	$Pnma$	47.6
47268	$C2/c$	75.2
48747	$C2/m$	47.6
59960	$P2_1/m$	49.6
82151	$Pmn2_1$	80.9

Initially, we also tried to refine the PDF data against all possible known vanadium oxides in the ICSD, ranging from V_2O (V^+ , ICSD 30415) to V_2O_5 (V^{5+} , ICSD 60767), as well as all other vanadium oxides with intermediate oxidation states, including mixed valence compounds, such as V_3O_7 (ICSD 2338) or V_6O_{13} (ICSD 62117). Eventually, the only structural model that matched our PDF data was V_4O_9 (ICSD 15041) by providing a good fit while maintaining physical values for interatomic V-O and V-V distances and atomic displacement parameters. Additionally, following Reviewer 3's suggestions (see Reviewer 3, comment 17), we were able to identify a second phase in **1-200**. This phase is the ammonium vanadium bronze, $(NH_4)V_4O_{10}$. The introduction of this phase into the refinement improved the fit considerably from $R_w = 20.4\%$ ($V_4O_9 + 1$) to $R_w = 14.8\%$. Our results indicate that **1-200** is mainly composed of these three phases in an approximate weight fraction $V_4O_9:(NH_4)V_4O_{10}:\mathbf{1}$ 50(5):35(4):15(7). Therefore, we can conclude that the

thermal decomposition of **1** undergoes transformation to form intermediate mixed-valence V_4O_9 and $(NH_4)V_4O_{10}$ structures before its full conversion to V_2O_5 beyond 250 °C.

Besides, the colouration of the intermediate is attributed to IVCT. Could this be proven in the UV-Vis absorption spectra?

V(IV)/V(V) IVCT is well known in molecular species, typically giving a broad signal across the visible region, with examples often centred ~900 nm (e.g. 500-1400 nm has been reported) e.g. see Streb et al *Angew Chem*, 2025. Further to this localised V(IV) d-d transitions are also possible and typically fall in the visible region (e.g. $VO(acac)_2$ absorbs has broad signals at ~580 and 700 nm) e.g. *J. Mater. Chem. A*, 2024, 12, 26645–26666.

Considering the strong absorbance of the material across the UV/visible region, we have repeated the diffuse reflectance absorption spectra by diluting the black (**1_200**) phase in $BaSO_4$ to record a spectrum at 10 wt%. This spectrum (now in Fig S11) only shows a broad absorption across the entire region, with very slightly higher absorption in the near IR region.

We attribute the absorption spectra to a combination of d-d transitions and broad IVCT absorption.

9. In the electrochemical testing, cycling shows increasing capacity up to 400 mAh g^{-1} . Could the authors clarify the mechanism (structural transformation vs activation)? Are there ex-situ characterisation data after cycling to support this? How reproducible are the cycling data across samples?

We thank the reviewer for raising this important point. The electrochemical data presented in our manuscript were obtained from two independent samples, which exhibit consistent behaviour. Nonetheless, we acknowledge that small variations in heating rate or temperature between different ovens or furnaces can influence the relative $V_4O_9:(NH_4)V_4O_{10}:1$ ratio in the resulting products, which may in turn lead to differences in the resulting voltage curves.

Figure S55 – Ex-situ PXRD ($Cu K\alpha$) of **1-200** electrodes before (black) and after 10 cycles (red) at 100 mA g^{-1} between 1.0 – 4.3 V. Sharp peaks marked with * are due to the Al current collector.

To investigate possible structural transformations during cycling, we collected ex-situ powder X-ray diffraction (PXRD) data on samples before and after 10 galvanostatic cycles at 100 mA g^{-1} . The ex-situ PXRD data suggests structural changes have taken place upon cycling (**Figure S55**). The ex-situ PXRD patterns indicate that structural changes occur

Figure S52. Representative voltage profile (top) and specific capacity *versus* cycles (middle) and differential capacity plots (bottom) of 1 heated to 200°C cycled at $20 \text{ mA}\cdot\text{g}^{-1}$ in a Li-ion half-cell using Li metal as a counter electrode and $1\text{M Li}[\text{PF}_6]$ in EC:DMC 1:1 (v/v). The sudden capacity increase after the 5th cycle is due to the change in voltage window from 2.0 – 4.3 V to 1.0 – 4.3 V.

upon cycling, however, the amorphous nature of **1-200** and the absence of well-defined Bragg peaks prevent us from resolving the details of this structural change. To address these questions more reliably, dedicated *in-situ* or *operando* measurements involving local characterisation techniques –such as pair distribution function analysis or X-ray absorption spectroscopy– would be needed, which are better suited to probing local structural rearrangements in amorphous materials.

These structural changes are, however, much more evident in the differential capacity curves from the data cycled at 20 mA g^{-1} . This data shows drastic changes in the voltage profile after extending the voltage window to 1.0 – 4.3 V, with new oxidation features appearing on charge around 2.0 – 2.5 V and between 2.7 – 3.5 V whereas a pronounced reduction feature appears below 2 V on discharge. This latter contribution corresponds to the main discharge plateau and accounts for approximately 66% of the total discharge capacity. Conversely, the discharge plateau centred at 2.6 V disappears after the 5th cycle. With continued cycling, these features gradually shift to higher potentials, accompanied by an increase in capacity.

These observations are consistent with progressive, irreversible structural changes in **1-200**. Given that the shape of the voltage curves is related to the energetics of lithium intercalation/de-intercalation, any sustained variations of the electrochemical profile are typically indicative of structural changes. This is also supported by the extensive literature reports on vanadium oxides, which are known to undergo multiple phase transitions during lithium intercalation (e.g. see refs 50, 85). Overall, the gradual increase in capacity to $\sim 400 \text{ mAh g}^{-1}$ is most consistent with progressive structural transformations rather than from electrochemical activation.

To clarify this point, we have included the differential capacity plots in the supplementary information and modified the main text accordingly.

10. The manuscript frames BiVO_4 largely as a photoanode material, but the new $\beta\text{-BiVO}_4$ phase is only structurally described. Could the authors better contextualise how this discovery fits into the broader landscape of polymorph engineering for photocatalysis? For example, can this be extended to stabilising other metastable polymorphs via SSP routes? Similarly, how does this work compare with other recent reports on metastable BiVO_4 phases?

The following section is added to the introduction, with additional references.

“However, recent reports of high performance photoanode materials show that Mo-doping alters phase stability and promotes tetragonal scheelite,²⁸ and, whilst the tetragonal zircon phase has a higher band-gap, it is of interest for forming type-II heterojunctions due to its high crystal symmetry,²⁹ or can be used as a photocathode,³⁰ therefore, polymorph engineering of BiVO_4 is of wide interest.”

A sentence is added to the conclusions to emphasise the possibilities of finding further metastable phases by using SSPs, and a manuscript by Ceder et al discussing solid state preparation of metastable phases (Sci Adv, 2024) is now referenced in the paper.

Reviewer #2 (Remarks to the Author):

The authors report very interesting results of thermal decomposition reactions using mono-, bi- and trimetallic polyoxovanadates (POVs). Upon heating, all three compounds lose volatile species and amorphous intermediates are formed. Increasing the temperature crystalline V₂O₅ and BiVO₄ developed and a new kinetically stabilized polymorph of BiVO₄ was identified, which is named β -BiVO₄. A variety of methods were applied to shed light on local structures of the intermediate amorphous phases. Final crystalline materials were characterized with Rietveld refinements of XRD data. State-of-the-art methods include PDF analysis, in-situ X-ray scattering, solid state NMR, EPR, chemical analyses, UV-Vis, FT-IR spectroscopy, XANES, TGA-MS, and DSC. At the end, the authors were able to develop a comprehensive picture about the amorphous phases and to identify the new polymorph of BiVO₄. The investigations, experiments and interpretation of the data go far beyond what the reviewer has seen in literature.

We thank the reviewer for their time assessing the manuscript and for their supportive comments.

Before the very well written manuscript can be accepted a revision is required. First a personal remark of the reviewer: the part dealing with the results of Li intercalation/insertion into 1-200 is out of place and does not add information about the main aim of the manuscript: understand phase formation during thermal decomposition, identify local and long-range structures related to chemical compositions, identify new crystalline compounds. But it is the decision of the authors whether they still want to include this part in a revised version. In this case much more information about the electrochemical experiments must be supplied.

We thank the referee for raising this issue and apologise for failing to provide key experimental information on the electrode and coin cell preparation for electrochemical testing. We have included that information now within the Methods section in the main text, which now reads as:

Electrochemical measurements – 1-200 coatings for electrochemical cell testing were prepared by slurry casting onto an Al foil. The slurry was obtained by mixing 80 wt.% of active material (AM), 10 wt.% of Super C65 Carbon black, and 10 wt.% of polyvinylidene fluoride (PVDF, Merck) as binder with N-methyl-2-pyrrolidone. The resulting coating was dried at 110 °C overnight in a vacuum oven before use. Electrochemical tests were performed by preparing CR2032-type coin half cells using the coated cathode (14mm diameter, mass loading: 2-3 mg cm²), Li metal anode (15 mm, Merck), Whatman glass microfiber separator (18 mm) and 1.0 M LiPF₆ in EC:EMC (ethylene carbonate/methyl carbonate, 3:7 V/V), which were assembled inside an Ar-filled glovebox. Galvanostatic cycling tests were performed in the voltage ranges of 2.0 – 4.3 V and 1.0 and 4.3 V at current densities of 20 mA·g⁻¹ and 100 mA·g⁻¹.

We would also like to note that, on reflection, we have re-ordered the manuscript to include the electrochemical study at the end, we feel this strengthens the narrative in the first section, but retains, what we believe are, interesting electrochemical studies.

I. 50: the thermal decomposition of ammonium decavanadate hexahydrate was investigated by Todorović, Mioč, Holclajtner-Antunović and D. Šegan; please add the reference

Thank you for this reference, it has been included.

I. 85: please add the chemical formula of compound 2.

This has been added

I. 100/101: if the sample is a 1:1 crystalline phase mixture one would expect that the PXRD patterns shows reflections of both components. Fig. S2 shows only the calculated pattern of one of the two components. A comment on this is required. If possible please rescale the calculated pattern.

Thank you for raising this question. The PXRD pattern is a very good match for the reference pattern (which is based on the crystal structure with 6 ammonium cations). The elemental analysis is also consistent (within expected error) with the formula with six ammonium cations, therefore, on balance, we have removed discussion of the protonated form (with 5 ammoniums) from the main text but retain a note to discuss this as a possibility in the supporting information. The calculated pattern is rescaled on Fig S2.

I. 111: ... Thermogravimetric analysis (TGA-MS/DSC) shows several consecutive mass...In Fig. S3 the DTG curves should be included helping to identify the distinct mass loss steps.

This is now included in the SI.

Fig. S4: please include an arrow showing the endo-exo direction. At least in the SI a short discussion of what is seen in the DSC curves should be included. What is the sharp event occurring just below 700 °C in the DSC curve recorded in air?

An arrow is included. The sharp endothermic peak is from melting of V₂O₅, this is added to the FigS4 caption along with expanded descriptive comments to accompany the plot.

Assignment of emitted masses: is really atomic oxygen emitted or is it NH₂?

This is a good point, as we are not certain, both possibilities are now listed.

Fig. S5: why were different heating rates chosen for the experiments in air and N₂?

These were standard experimental settings, we do not anticipate a major change in the results based on the change in heating rate (see similar TGA plots).

Fig. S7: please add a Table summarizing results of the Rietveld refinements.

In compliance with the reviewer's suggestions, we have added a table (Table S3) to complement figure S7 with a summary of the results from the Rietveld fits at different temperatures.

Table S3 – Summary of the results from the Rietveld refinements of **1** calcined at different temperatures (Figure S7).

Temperature / °C	a / Å	b / Å	c / Å	V / Å ³	Rwp / %
250	11.5518(4)	3.5750(1)	4.4409(3)	183.40(1)	1.32
300	11.5543(2)	3.57487(7)	4.4461(1)	183.65(1)	0.80
350	11.5564(2)	3.57417(7)	4.4559(1)	184.05(1)	0.62
400	11.5588(2)	3.57395(7)	4.4664(1)	184.51(1)	0.50
450	11.5626(2)	3.57363(6)	4.4777(1)	185.02(1)	0.42
500	11.5664(2)	3.57354(5)	4.4887(1)	185.53(1)	0.36
550	11.5724(2)	3.57337(5)	4.5001(1)	186.09(1)	0.33
600	11.5779(2)	3.57321(5)	4.5113(1)	186.63(1)	0.32
30	11.5505(2)	3.57542(7)	4.3903(1)	181.31(1)	0.31

I. 118-121: the reviewer suggests performing the analyses on a sample obtained at identical temperature. Here XRPD was measured on a samples obtained at 160 °C, IR on a sample heated to 175 °C and H, N analysis was done on a sample heated to 150 °C. This is somewhat disturbing.

The partially crystalline phase at 160°C forms in a small temperature window (as not observed at 140°C or 180°C). IR data for 150°C and 175°C are included in Fig S8, and these are broadly similar, therefore we expect these spectra to be consistent through this temperature window (noting that at 150°C some contribution from starting material is also present). H, N analysis at the same temperature provides a consistent picture, suggesting a combination of (dehydrated) **1** and [NH₄]₂[V₆O₁₆]. The text has been restructured and amended to improve this discussion.

I. 125: ... powdered **1** was estimated to have average crystallite sizes of ~10 nm...at what temperature?

'Room temperature' is added to text

I. 129: is it not possible estimating the bandgaps from the UV-Vis data (Kubelka-Munk or tauc plots)? (see also Fig. S27). Why are no UV-Vis data are provided for 3-Zn and the decomposition products?

The absorption onset for **1** and **2** (by Tauc's analysis) are now included in the caption of Fig S11 and Fig S27.

The UV spectrum of **3-Zn** is previously published in Adv Mater, 2018, 1804033, and is similar to **2**. Photographs of the decomposition of **3-Zn** are included in Fig S28 and show the expected colours.

I. 130:... ruling out carbon contamination....where should the carbon come from? 1 contains no carbon.

We have rephrased this sentence to make sure that it is not misleading.

I. 139: The presence of V(IV) was confirmed by EPR (Fig S13). Is it not possible estimating the amount V(IV) from the spectrum?

Accurate spin counting experiments are challenging, and we were not able to measure this in this instance.

I. 217: ... noteworthy that precipitation of **2** may generate an amorphous form (2amor) which...Here (or in the SI) the procedure for obtaining 2amor should be described (the reviewer has not found information about this)

Thank you for noticing this, a note to explain this has been added to the supporting information.

p. 10: the authors recoded NMR spectra of **2**, on p. 11 TGA was performed on 2amor and the authors write that 520 °C is required to complete the mass loss for **2**. This is a little bit confusing. Please clarify.

Thank you for noting this, we have amended the text to be 2amor where appropriate.

I. 231: ... decomposition (and water is also lost from **2**, Fig S18-26)...Fig. S18 displays the PXRD pattern of **2**

Thank you, this is corrected

I. 236: ... indication of CO₂, H₂CO, SO₂ and H₂S loss occurring...In Fig. 23 no mass is indicated for H₂S

changed to 'indication of CO₂ and H₂CO, and some evidence of SO₂ and/or H₂S loss occurring'

The DSC curves (Fig. S22) should be discussed at least in the SI because significant differences are seen in the traces for air and N₂.

Further information is added to the SI captions and a note to main text to link the mass gain in the TGA (under air) with an exothermic oxidation reaction shown in the DSC.

I. 266, Fig. 6: please adjust the chemical formula and the cubic space group

Subscripts added, thank you for spotting these.

I. 292: it is a lone electron pair

corrected

I. 322/323: one digit is enough for the estimated uncertainty/standard deviation

corrected

I. 326: Table S8

capitalised

I. 338: 300 °C instead of deg C

corrected

I. 340-351: please indicate what polymorph of BiVO₄ is meant

m-BiVO₄ added

I. 354/355: beta-BiVO₄ instead of b-BiVO₄

corrected

Fig. S19: there are remarkable intensity differences between experimental and calculated XRD patterns for 3-Zn. A comment is required.

A comment is added to the SI caption

"Powder XRD pattern of **3-Zn** (at room temperature) compared to a simulated pattern from the single crystal structure (at 100 K), and pattern after heating *ex-situ* to 60 and 175°C (collected at room temperature). The PXRD data for **3-Zn** shows good agreement with the simulated pattern with minor differences likely due to the different

temperatures of collection, and the imperfect single crystal model which exhibits significant disorder of co-crystallised solvent molecules.”

Fig. S4 and S22: include an arrow showing the endo-exo direction

This has been done

Fig. S23: the assignment of the masses should be reconsidered. 2amor does not contain nitrogen. See also Fig. S26; what is the dip in the mass traces at around 300 °C?

We assume the reasonably strong signal at $m/z = 29$ is protonated N_2 forming in the gas phase.

The TGA-MS data all suggest lesser emission of gasses at higher temperatures, presumably due to most ligand decomposition occurring at lower temperatures as evidenced in the elemental analysis data.

Fig. S24: please show also the DTG curves.

Added

Fig. S44: It seems that the figure caption contains duplicate information.

The caption is ammended to avoid repetition.

In the Method section the authors should mention that crystals with composition $[VO(DMSO)_5][NO_3]_2$ and 3-VO were obtained and the crystal structures were determined, more information is given in the see SI.

Thank you for this suggestion, this has been added

We are extremely grateful for the detailed analysis throughout the paper by this reviewer.

Reviewer #3 (Remarks to the Author):

The authors report on different single source precursors containing V, V+Bi or V+Bi+Zn, which they thermally transform to intermediate phases and final oxides. They follow the transformation by ex situ and in situ characterization methods, which also enable the study of the atomic structures of amorphous and disordered phases. The aim of the work was to obtain multi-element solids directly from the precursor.

We thank the reviewer for the care and attention in reading the article and for their constructive input.

In case it was the goal of the studies to synthesize one-structure multi-metal oxides, it failed e.g. as the 3-Zn-V-Bi system formed phase mixtures.

The project goes far beyond this aim, with a very detailed investigation of a molecule to material transformation in a complex system using advanced techniques. Highlighting how this approach can yield new phases and unexplored materials science. It has already been reported that 3-Zn transforms to a combination of Zn-doped BiVO₄ and V₂O₅, and in the article we discuss whether this could in fact be beneficial.

The authors also tested the electrochemical properties of the vanadium system, but the results are not discussed in relation to literature reports on the same composition (e.g., DOI <https://doi.org/10.1039/D1QI00747E>).

We thank the reviewer for this suggestion. In this particular case, V₄O₉ is used in aqueous Zn-ion batteries and it is, thus, not directly comparable. However, we note there is a similar behaviour in both systems, displaying both a specific capacity increase with consecutive cycles. We would like to note that, although the authors in this manuscript first ascribe this phenomenon to structure activation, it is evident from the diffraction data (Figures S4 and S9 in the referenced article) that the structure of V₄O₉ changes, losing its long-range order in the first few cycles, during which the capacity increase is observed. We would also like to note that other similar capacity increase have been shown in VO_x nanomaterials in Li-ion batteries (e.g. see *Electrochem. Solid-State Lett.* **4**, A129 (2001)). Additionally, previous reports on the use of V₄O₉ and the vanadium bronze (NH₄)V₄O₁₀ as Li-ion battery cathodes (See response to reviewer 3, comment 17) have shown large structural changes upon Li intercalation. Therefore, it is expected that similar structural changes may be happening in 1-200 upon discharge.

In compliance with the reviewer's suggestion and to add more context to the electrochemistry results, we have added the suggested reference, and additional references on the use of V₄O₉ and the vanadium bronze (NH₄)V₄O₁₀ (See response to reviewer 3, comment 17) as Li-ion battery cathode to the discussion, which now reads as:

“A similar behaviour has previously been reported for V₄O₉ in aqueous Zn-ion batteries, where structure amorphization has been observed upon Zn²⁺ intercalation alongside a capacity increase during the first 100 cycles.⁸⁶ Additionally, previous studies of (NH₄)V₄O₁₀ and V₄O₉ in Li-ion battery cathodes have shown significant structural changes upon lithium intercalation with V₄O₉ irreversibly transforming into a disordered rock salt structure.^{59, 62, 87, 88} The observed specific capacities of 1-200 are in accordance with the reported V₄O₉, which can intercalate up to ~6 Li⁺ (~3.7-4 Li⁺ when cycled between 1.8 – 3.5 V),⁶² and with other reported mixed-valence vanadium oxides nanomaterials, which can incorporate multiple Li per formula unit and whose structure has been shown to change upon electrochemical cycling.⁸⁵”

For compounds 2 and 3-Zn, the manuscript provides some insights into the crystallization pathways, but the structure-property relationship was not investigated. The chapters on sample 1 and on sample 2/3-Zn are therefore not really related. The authors claim to be able to adjust the oxidation state of vanadium through synthesis. In

my opinion, however, this does not justify publication in Nature Communications. For these reasons, I recommend submission to a more specialized journal.

We have restructured the article to maintain a focus on the detailed analysis of molecule to material transformations, the electrochemical study of **1_200°C** has been moved to the end of the article.

General comments:

1. The authors apply a variety of techniques, which sometimes makes the manuscript difficult to follow. The manuscript could be more structured to allow identification of the differences and similarities between the different systems.

We appreciate the concern of the reviewer, however, we have spent a long time considering the best way to present the large quantity of characterisation information necessary within this manuscript and we believe that we have achieved this to the best of our ability in its current form. We note that other reviewers comment that the manuscript is well-structured.

2. The quality of the experimental data varies significantly. For example, Figure S19 plots all XRD patterns for different 2θ ranges.

The majority of PXRD information for molecular compounds with large units cells is captured at the lower 2θ range (as is evident in the predicted patten for 3-Zn in Fig S19). Therefore, when analysing this compound a lower 2θ range was chosen. This does not affect the quality of the data.

3. The authors also provide a large number of figures in the supplementary information (SI). Nevertheless, they should be more consistent in terms of representation. For example, some figures use ' 2θ /(degrees)' and others use "Two theta ($^\circ$) or 2θ / $^\circ$ ".

All PXRD figures now presented with consistent axis labelling

4. The figures have different fonts used for labelling the axes, etc. This makes the manuscript appear inconsistent. It looks as if different authors have contributed different measurements and everything has simply been put together in one file.

Whilst we appreciate the reviewer's attention to detail, this large collaborative project was indeed conducted by a variety of researchers, across several institutions, working on different techniques which sometimes required different software. If anything is unclear, we would be very happy to amend it appropriately.

5. XRPD in Figures 2 and 6a: The details are difficult to discern. The details that could indicate intermediate phases are not visible. Such figures are not helpful at all. I

recommend adding the Rietveld refinement plots as individual plots to the Supporting Information, scaled reasonably to allow the difference curves and simulations to be seen. The pattern should appear as shown in Figure S48.

We understand the reviewer's concern regarding the figures. The figures in the main manuscript show the broad changes in PXRD pattern, and whilst we appreciate that the fine details are hard to observe, they provide an overarching story to the reader. We have modified the plot in Figure 6a to make it easier to read and have added a new figure in the ESI containing three representative fits at 380, 420 and 500 °C (Please note that the fit at 400 °C is already included in Figure 6c). The new plots are shown below:

Figure S51 – Representative Rietveld fits of the thermal transformation of 3-Zn at 380, 420 and 500 °C.

6. The discussion of the ex situ and in situ XRD patterns is very difficult to follow, especially for compounds 2 and 3Zn. One has to jump back and forth between the main manuscript and the SI.

Due to the significant quantity of data collected, we are unable to show everything in the main text and we believe the layout and cross refereeing to the SI are appropriate.

Figure 6a – New figure 6a in the main text showing the Rietveld refinements during thermal transformation of **3-Zn**.

7. For clarity, I would plot the XRDs of the starting materials together with the calcined samples.

Thank you for this suggestion. This has already been done in Fig 2. (1) & Fig S19 (3-Zn). We have now also added this to Fig S30 for 2 and added further data to Fig S19.

8. Are the ex situ and in situ phases the same for the given temperatures? Do reversible HT phases transform back to LT phases during cooling? I think this will be addressed later but it would be helpful to reconsider the structure of the discussion.

The discussion of high temperature *ts*-BiVO₄ and its transition to *m*-BiVO₄ at room temperature is rephrased for additional clarity. There are minor changes between ex-situ and in-situ data but note that this more likely reflects differences in crystallinity in 2 and 2^{amorph} as *ts*-BiVO₄ begins to crystallise at an earlier temperature from the amorphous precursor.

9. It is also difficult to follow the phase transformation pathways when samples 2 and 3-Zn are discussed in the same sentence.

We appreciate the reviewers concern and have altered some parts of the text to improve clarity.

10. Explanation of 2 or 3-M is missing

A full formula definition for 3-M is added at first usage for additional clarity

11. Crystal structure data used for XRD and PDF refinements (ICSD numbers etc)

We apologise for failing to provide this information. The ICSD numbers of the phases used for Rietveld refinement for V_2O_5 , $ms\text{-BiVO}_4$, $t\text{-BiVO}_4$ are 60767, 100602, and 144629, respectively. Besides the above, for the PDF refinements, we also used the following phases with their respective ICSD codes: V_2O_9 (ICSD 15041), $(NH_4)V_4O_{10}$ (ICSD 230493) and $(NH_4)_6V_{10}O_{28}$ (ICSD 161217). We have now also included this information in the methods section and in the supplementary information.

12. A PDF of the starting compound would be helpful for following the transformations.

We thank the reviewer for the suggestion. We have collected and added the PDF of the precursor 1, alongside those of 1-200 and 1-400, as a new figure in the ESI (See **figure S12** below) for a direct comparison. Our results show the complete loss of features of 1 upon calcination at 200 °C, suggesting its complete or almost complete transformation.

Figure S12 – Comparison of experimental PDFs of 1 before (blue) and after calcination at 200 (black) and 40 °C (orange). $Q_{\max} = 22.0 \text{ \AA}^{-1}$.

13. PDF: Page 6, line 134: “1-200°C reveals a different PDF pattern, which was found to resemble V_4O_9 (Pnma,53, 54 $R_w = 30.4\%$ after refining thermal factors and atomic positions, Fig S12).”

Comment: please provide a table showing the refined U_{iso} values, lattice parameters, and refined atomic positions, including the resulting main interatomic distances.

In compliance with the reviewer suggestion, we have added a table containing the refined atomic coordinates and U_{iso} values as a table in the ESI (see table below) as well as a new Figure to show the highlight the contributions from V-O and V-V correlations from the fit of this phase (Figure S13 middle).

Table – Structural parameters from the PDF refinements of 1-200 using the V_4O_9 structural model ($Pnma$, ICSD 15041). Refined cell parameters: $a = 18.58(7) \text{ \AA}$, $b = 3.578(8) \text{ \AA}$, $c = 9.52(1) \text{ \AA}$, $\alpha = \beta = \gamma = 90^\circ$.

Atom Label	x	y	z	Occupancy	$U_{iso} / \text{\AA}^2$
V1	0.313(3)	0.25000	0.434(6)	1	0.012(4)
V2	0.160(2)	0.25000	0.212(5)	1	0.012(4)
V3	0.990(3)	0.25000	0.231(5)	1	0.012(4)
V4	0.067(2)	0.25000	0.523(5)	1	0.012(4)
O1	0.025(5)	0.75000	0.053(1)	1	0.007(5)
O2	0.032(7)	0.25000	0.09(1)	1	0.007(5)
O3	0.393(6)	0.25000	0.35(1)	1	0.007(5)
O4	0.151(6)	0.75000	0.18(1)	1	0.007(5)
O5	0.290(7)	0.75000	0.51(1)	1	0.007(5)
O6	0.250(6)	0.25000	0.32(1)	1	0.007(5)
O7	0.160(6)	0.25000	0.60(1)	1	0.007(5)
O8	0.963(6)	0.75000	0.26(1)	1	0.007(5)
O9	0.095(6)	0.25000	0.36(1)	1	0.007(5)

Figure S13 middle – Comparison of experimental and simulated PDF for 1-200 highlighting contributions of the main interatomic V-O and V-V correlations.

We would also like to note that, following the suggestions raised by other reviewers (see Reviewer 3, comment 17), we also noticed the presence of a third phase in the PDF of 1-200, which resembles the structure of the vanadium bronze $(NH_4)V_4O_{10}$ (ICSD 230493). The inclusion of this phase significantly improved the fit from $R_w = 20.4\%$ ($V_4O_9 + 1$) to $R_w = 14.8\%$. This suggests that **1-200** is mainly composed of these three phases in an approximate weight fraction $V_4O_9:(NH_4)V_4O_{10}:1$ 50(5):35(4):15(7), suggesting that the thermal decomposition of **1** undergoes transformation to form intermediate mixed-valence V_4O_9 and $(NH_4)V_4O_{10}$ structures before its full conversion to V_2O_5 beyond 250°C .

14. PDF: Page 6, line 135: "The fit was improved by adding a contribution from either decavanadate or hexavanadate anions ($R_w = 20.1\%$, (1) = 21.4 wt%, Fig 3b; or $R_w = 26.7\%$, $([NH_4]_2[V_6O_{16}]) = 13.2$ wt% respectively),..."

Comment: Which structure was included? To my opinion, the refinement plots in Figures 3b and S12 do not differ significantly.

We apologise for the misunderstanding. The fit in Figure 3b was obtained after including precursor **1**, which is the one that gave the best fit. We understand the reviewer's concern about visual inspection of both fits not revealing very clear differences between them. However, there is a substantial reduction in the goodness-of-fit parameter (R_w , calculated as $R_w = \sqrt{\left(\frac{\sum(G_{obs}(r_i) - G_{calc}(r_i))^2}{\sum G_{obs}(r_i)}\right)}$) from 30.4 to 20.1 % after including the structure of **1**, which clearly indicates a significant improvement in the fit of PDF data.

15. Page 6, line 139: "The presence of V(IV) was confirmed by EPR (Fig S13). 1-200°C was dissolved in 2M H₂SO₄ and a double titration method used to identify V(IV) content,⁵⁵ which indicated only 2.7 ± 1 % of V was V(IV). V₂O₅, **1** and 1-200°C also appear almost identical by X-ray Absorption Near Edge Structure (XANES) analysis, consistent with a majority of V(V) in 1-200°C (Fig S14)."

Comment: EPR confirms the presence of V⁴⁺, but XAS shows that mainly V⁵⁺ is present.

Yes, only a small V(IV) content is present therefore this is not picked up in the XANES analysis. A comment is added to clarify that XANES is less sensitive to detecting V(IV) compared to EPR spectroscopy.

16. How do these results compare with those of the PDF, which shows a mixture of V⁴⁺ and V⁵⁺ for V₄O₉? The authors discuss this later, proposing that V⁴⁺ be replaced with V⁵⁺. However, perhaps this is too simple an explanation.

We thank the reviewer for bringing out this point. From the PDF fit alone, we cannot confirm or rule out the presence of V⁴⁺ in the structure. Additionally, the X-rays are not sensitive to changes in oxidation state and the V-O bond distances in **1-200°C** and **1** after heating to 400°C are too similar to see any clear differences in bond distance due to different oxidation states.

Changes to the discussion are made in light of the inclusion of (NH₄)V₄O₁₀ phase and comparison of the expected and determined V(IV) content from these phases. The new text is included below:

"From this mixture of materials, ~35% of V might be expected in the V(IV) oxidation state and the presence of V(IV) was confirmed by EPR (Fig S15). However, **1-200°C** was dissolved in 2M H₂SO₄ and a double titration method used to identify the amount of V(IV) content,⁶¹ and a lower value of 2.7 ± 1 % V(IV) was identified. X-ray Absorption Near Edge Structure (XANES) analysis confirmed that V(V) is dominant as the spectra of V₂O₅, **1** and **1-200°C**

appear almost identical, (noting that XANES is less sensitive to the presence of V(IV) than EPR spectroscopy, Fig S16). This low V(IV) content is consistent with previous studies,³⁵ yet is surprising considering the black colour of **1-200°C**.

The reduction of some V at intermediary stages of heating is caused by a redox reaction with ammonia evolved from **1**, in fact, treating V₂O₅ with NH₃ is a synthetic route to V₄O₉.⁶² Partial reduction is consistent with literature reports,^{32, 34, 35, 39} but here we extend on these by identifying a likely structure of this initial amorphous phase. In **1-200°C** the local connectivity resembles a mixture of V₄O₉, and (NH₄)V₄O₁₀ but the low V(IV) content implies that most sites are occupied by V(V), implying that these may be oxygen-rich kinetically stabilized phases."

17. Page 7, line 153: "In 1-200°C the local connectivity resembles V₄O₉ but the low V(IV) content implies that most sites are occupied by V(V), implying that this may be a kinetically stabilized phase. In related solution studies, heating [NMe₄][H₂V₁₀O₂₈] under hydrothermal conditions is reported to form mixed valence [NMe₄][V(V)₃V(IV)O₁₀] with a layered structure.^{50, 57}"

Comment: Why have the authors not simulated this structure?

We thank the reviewer for this suggestion. We have now tested the (NH₄)V₄O₁₀ structure (ICSD 230493, *Chem. Mater.* **30**, 3690–3696 (2018)), which has also been tested as Li-ion battery cathode. Interestingly, although using the (NH₄)V₄O₁₀ alone resulted in a worse fit (Rw = 46.6%) than V₄O₉ alone (Rw = 30.4%) the combination of V₄O₉ with (NH₄)V₄O₁₀ does provide a better fit to the experimental PDF (Rw = 17.8 %, Relative mass fraction: 55(8):45(8), V₄O₉:V₄O₁₀), than the fit containing V₄O₉ + precursor **1**, while maintaining physically meaningful interatomic distances and atomic displacement parameter values. However, this also comes at the expense of adding a higher number of parameters to the refinement. The best fit was obtained after also including precursor **1** in the refinement, with Rw = 14.8%, for which only cell parameters and atomic displacement parameters were refined. In this case, given the similar value of atomic displacement parameters obtained for V and O (U_{iso}(V) = 0.009(3), U_{iso}(O) = 0.009(6)), we constrained them to be the same to reduce the number of parameters. The relative mass fractions obtained from the refinement are V₄O₉:V₄O₁₀:**1** 50(5):35(4):15(7), also confirming that V₄O₉ is the majority phase. We would also like to note that adding (NH₄)₂V₆O₁₆ phase at this stage resulted in negative scale factors and unphysical parameters for this phase, ruling out its presence. From this analysis, we can conclude that decomposition of **1** at 200°C likely forms a mixture of V₄O₉ and V₄O₁₀ phases, before it's complete transformation to V₂O₅ beyond 250 °C. The PDF fits are shown in Fig S14 highlighting the contribution of each phase. These have also been discussed in the main text.

18. Electrochemical measurements suggest the presence of other phases in the sample calcined at 200 °C, such as V₆O₁₃, but confirmation of these phases is missing.

We apologise for the misunderstanding. Although we initially had compared the electrochemical behaviour of **1-200°C** to that observed in other mixed valence vanadium oxides, such as V_6O_{13} , the electrochemical data does not suggest in itself the presence of V_6O_{13} . We understand that this may be misleading and have, therefore, rewritten the statement as:

[...]. A similar behaviour has previously been reported for V_4O_9 in aqueous Zn-ion batteries, where structure amorphization has been observed upon Zn^{2+} intercalation alongside a capacity increase during the first 100 cycles. Additionally, previous studies of $(NH_4)V_4O_{10}$ and V_4O_9 in Li-ion battery cathodes have shown significant structural changes upon lithium intercalation with V_4O_9 irreversibly transforming into a disordered rock salt structure. The observed specific capacities of **1-200°C** are in accordance with the reported V_4O_9 , which can intercalate up to $\sim 6 Li^+$ ($\sim 3.7-4 Li^+$ when cycled between 1.8 – 3.5 V), and with other reported mixed-valence vanadium oxides nanomaterials, which can incorporate multiple Li per formula unit and whose structure has been shown to change upon electrochemical cycling.⁶⁵ [...]

On reflection, we believe it is more accurate to compare the behaviour of **1-200°C** to other mixed-valence vanadium oxide nanomaterials, where it has been seen as similar increase in capacity during the first few cycles. Nonetheless, we note that this increase in capacity might also be influenced by the presence of unreacted precursor in **1-200°C**, as suggested by PDF analysis.

19. The thermally induced structure transformations for sample 1 are not convincingly supported by experimental PDF data, which show only V_4O_9 .

We respectfully disagree with the reviewer. Our intention was never to claim the sole formation of V_4O_9 upon calcination of 1 at 200 °C. Initially, to characterise the structure of **1-200**, we tried to refine the PDF data against all possible known vanadium oxides in the ICSD, ranging from V_2O (V^+ , ICSD 30415) to V_2O_5 (V^{5+} , ICSD 60767, see also our response to comment 8 from reviewer 1), as well as all other vanadium oxides with intermediate oxidation states, including mixed valence compounds, such as V_3O_7 (ICSD 2338) or V_6O_{13} (ICSD 62117). Eventually, the only structural model that matched our PDF data was V_4O_9 (ICSD 15041) by providing a good fit while maintaining physical values for interatomic V-O and V-V distances and atomic displacement parameters.

Taking into account the reviewer's suggestion in the previous comment, we have also seen that there may also be another phase present, although in lower proportion to V_4O_9 (See response to comment 17). This phase could be modelled using the structural model for the ammonium vanadium bronze $(NH_4)V_4O_{10}$, and may account for the observed residual and N and H observed by elemental analysis of 1-200. As also noted in previous responses, we also show the possible presence of unreacted precursor 1 in our PDF analysis, indicating that V_4O_9 is not the only phase present.

20. Page 10, line 216: "...including the newly discovered structure [Bi₂V₁₂O₃₃Cl(DMSO)₆]₂[VO(DMSO)₅]₈(DMSO), 3-VO."
Comment: Reference for the crystal structure is missing.

This compound is reported in this paper for this first time. This information is included in the supporting information.

21. Page 14, line 286: " This gave excellent results, confirming that this new kinetic phase crystallizes in the cubic space group P213 with cell parameters $a = 6.9916(2)$ Å with Bi and V atoms sitting in the 3-fold symmetry axis of the cubic unit cell. Therefore, we dub this new phase β -BiVO₄."

Comment: it should be "cell parameter" instead of "cell parameters", as only one unit cell parameter is required for a cubic space group.

Corrected

22. An atom cannot "sit" "in" an axis, but it can be located on a symmetry element.

Thank you, this is corrected.

23. Page 14line 309: "2amor primarily yields tetragonal BiVO₄ (space group I41/a) at 300 °C followed by the subsequent crystallisation of V₂O₅ as the temperature increases (Fig S32), in contrast, for microcrystalline 3-Zn, V₂O₅ appears as the major species at 300°C before BiVO₄ phases grow in (Fig 6b)."

Comment: 2amor primarily yields tetragonal BiVO₄ (space group I41/a) at 300 °C followed by the subsequent crystallisation of V₂O₅ as the temperature increases (Fig S32). In contrast, for microcrystalline 3-Zn, V₂O₅ appears as the major species at 300°C before BiVO₄ crystallises (Fig 6b)."

Yes this is correct, see Fig 6b and Fig S32, this may be due to the influence of Zn in the BiVO₄ phases, causing disorder and requiring higher temperatures to crystallise. A comment is added.

24. Page 15, line 322: "For 2 after 400°C, PDF analysis suggests 42.8(55) wt% is β -BiVO₄, however this drops significantly by 420°C, at which point weight fractions are 43.4(37)% V₂O₅ : 52.6(35)% BiVO₄ : 4.0(18)% β -BiVO₄."

Comment: If the estimated standard deviation is larger than the refined main value (43.4(37%)), this should raise suspicions. Or was the standard deviation not used correctly (43.4(4))?

Standard deviations have been corrected, thank you for spotting this error.

25. For sample 1, electrochemical data were discussed, for samples 2 and 3-Zn, no data are obviously available. The entire discussion focuses on structure transformations. Again, this appears inconsistent with the first part.

We have restructured the manuscript to present the electrochemical findings at the end, as a separate section, so that the focus on structural transformations is clear.

26. The graphical presentation of the thermal pathways is somewhat puzzling. Usually, one would go from the bottom to the top, from blue to red, rather than downhill from the small crystals to the oxides.

The diagrams have been inverted

POLICIES AND FORMS REQUIRED FOR RESUBMISSION

* Please complete or update the following checklist(s) to verify compliance with our research ethics and data reporting standards. Address all points on the checklist, revising your manuscript in response to the points if needed.

The form(s) must be downloaded and completed in Adobe Reader rather than opened in a web browser. Each form must be uploaded as a Related Manuscript file at the time of resubmission.

Reporting summary:

This form has been completed, although most sections are not applicable to this study.

DATA AND CODE AVAILABILITY

* All Nature Communications manuscripts must include a "Data Availability" section after the Methods section but before the References. If any of the data can only be shared on request or are subject to restrictions, please specify the reasons and explain how, when, and by whom the data can be accessed. For more information on this policy and a list of examples, see:

<https://www.nature.com/documents/nr-data-availability-statements-data-citations.pdf>

This is in place

* We strongly encourage you to deposit all new data associated with the paper in a persistent repository where they can be freely and enduringly accessed. We recommend submitting the data to discipline-specific and community-recognised repositories; a list of repositories is provided here: <http://www.nature.com/sdata/policies/repositories>
Refer to our data policies here: <https://www.nature.com/nature-portfolio/editorial-policies/reporting-standards#availability-of-data>

A dataset has been created in the Warwick University research repository (WRAP) <https://wrap.warwick.ac.uk/194481/> and the raw NMR and XRD data are available on

Zenodo

* To maximise the reproducibility of research data, we ask that you provide a Source Data file containing the raw data underlying the following types of display items:

- Any reported means/averages in box plots, bar charts, and tables
- Dot plots/scatter plots, especially when there are overlapping points
- Line graphs
- Uncropped and unprocessed scans of all blots and gels including all quantified replicates. The edge of membranes, molecular weight ladders and loading controls should be presented on all blots. Where membranes have been cut, please ensure that at least one marker above and below is present. For an example of presentation of full scan blots, see the Source Data file of <https://www.nature.com/articles/s41467-020-16984-1#Sec35> and for more information, please refer to <https://www.nature.com/nature-research/editorial-policies/image-integrity>.

The data should be provided in a single Excel file with data for each figure/table in a separate sheet, or in multiple labelled files within a zipped folder. Name this file or folder 'Source Data', and include a brief description in your cover letter. The "Data Availability" section should also include the statement "Source Data are provided with this paper."

To learn more about our motivation behind this policy, please see: <https://www.nature.com/articles/s41467-018-06012-8>

A Source Data file is not necessary if all display items presented in the main manuscript and supplementary information can be reproduced from raw data and code that have already been shared in a public repository.

A dataset has been created in the Warwick University research repository (WRAP) <https://wrap.warwick.ac.uk/194481/> and the raw NMR and XRD data are available on Zenodo

ORCID

* Nature Communications is committed to improving transparency in authorship. As part of our efforts in this direction, we are now requesting that all authors identified as 'corresponding author' create and link their Open Researcher and Contributor Identifier (ORCID) with their account on the Manuscript Tracking System prior to acceptance. ORCID helps the scientific community achieve unambiguous attribution of all scholarly contributions.

You can create and link your ORCID from the home page of the Manuscript Tracking System by clicking on 'Modify my Springer Nature account' and following these instructions. Please also inform all co-authors that they can add their ORCIDs to their accounts and that they must do so prior to acceptance.

If you experience problems in linking your ORCID, please contact the Platform Support Helpdesk.

Email previously sent to all co-authors with instructions of how to do this